# Generative property enhancer: implicit guided generation through conditional density estimation

**Pedro O. Pinheiro**[1]   **Pan Kessel**[1]   **Aya A. Ismail**[2,*]   **Sai Pooja Mahajan**[1]
**Kyunghyun Cho**[1,3]   **Saeed Saremi**[1]   **Nataša Tagasovska**[1]

[1]Prescient Design, Genentech, [2]Guide Labs, [3]New York University

## Abstract

Generative modeling is increasingly important for data-driven computational design. Conventional approaches pair a generative model with a discriminative model to select or guide samples toward optimized designs. Yet discriminative models often struggle in data-scarce settings, common in scientific applications, and are unreliable in the tails of the distribution where optimal designs typically lie. We introduce generative property enhancer (GPE), an approach that implicitly guides generation by matching samples with lower property values to higher-value ones. Formulated as conditional density estimation, our framework defines a target distribution with improved properties, compelling the generative model to produce enhanced, diverse designs without auxiliary predictors. GPE is simple, scalable, end-to-end, modality-agnostic, and integrates seamlessly with diverse generative model architectures and losses. We demonstrate competitive empirical results on standard *in silico* offline (non-sequential) protein fitness optimization benchmarks. Finally, we propose iterative training on a combination of limited real data and self-generated synthetic data, enabling extrapolation beyond the original property ranges.

## 1   Introduction

Generative modeling has become an essential component to tackle computational design problems in scientific applications such as protein engineering, synthetic biology, material sciences or molecular design. An important application—the one we explore in this work—is the problem of *design optimization*[1], where the goal is to produce optimized samples (designs) according to some desired property(ies).

Conventional data-driven approaches are usually composed of two modules: (i) a generative model to propose design candidates, and (ii) a discriminative model to select optimized designs. The discriminative models are trained on available data to approximate the unknown objective landscape of the data. They are used to either rank and select generated samples or to guide the generation process toward samples with desired properties [1–18]. However, when data is scarce (usually the case in scientific applications), training a reliable discriminative model is often not feasible. To make the matter worse, we usually seek designs that lie in the tail of the distribution of the property which is typically very challenging to learn, making predictions even less reliable.

In this work, we challenge this paradigm by casting the problem of design optimization as a conditional generative modeling problem. We leverage the *implicit guidance* mechanism proposed by Tagasovska et al. [19] and take it a step further by going from an optimization perspective to a sampling-based one. This transition allows us to move beyond generating single point estimates and instead estimate

---

*This work was done while the author was at Genentech.

[1]Here, "design optimization" is taken in its broader, applied meaning common in science and engineering, different from its formal mathematical definition.

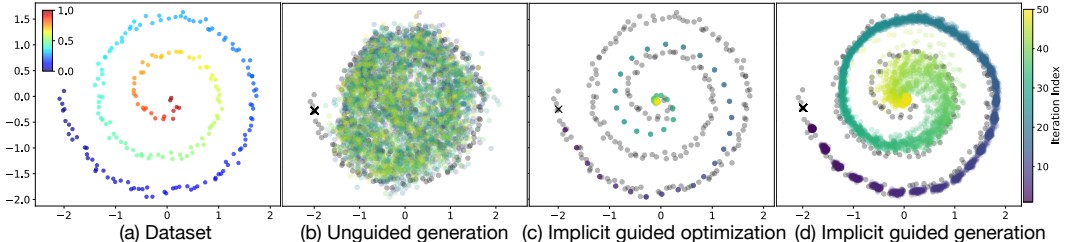

Figure 1: (a) An illustrative 2D dataset where the property increases counter clockwise. (b) Unconditional/unguided generative models trained on this data sample diverse points (marked as colored dots) but are unaware of their property values. (c) Implicit guided optimization [19] generates points with higher properties than the initial seed (marked as a cross), where each point is a subsequent optimization step. (d) Our method is able to sample diverse points with increasing values of the property. The colormap indicates the number of iterations. Appendix A further explores this example.

the probability density of improved designs. As a result, our framework is more general—readily applicable to diverse settings, data modalities, and generative models—and it mitigates a core limitation of point-based methods: their tendency to converge to local maxima [20, Section 4].

Our approach—*generative property enhancer (GPE)*—utilizes inherent pairwise relationships within the data: by matching lower-valued property samples $x$ directly to higher-valued samples $x'$, our method implicitly defines an *improved target distribution*. This "matching" strategy naturally forms a conditional density estimation problem wherein the generative model is compelled—via e.g. maximum likelihood training or approximating conditional score functions—to reproduce the distribution of samples with improved properties. Thus, excluding the reliance on any external discriminative model[2].

GPE provides an end-to-end, simple and scalable way to tackle design optimization problems with conditional generative models. The implicit guidance mechanism eliminates the need for an external predictor, reducing the complexity of the model and improving data efficiency. Moreover, it increases the effective training set size by taking into account pairs of points. Finally, our approach is general and can be applied to many different off-the-shelf state-of-the-art generative models and data modalities. Figure 1 shows how our approach works in an illustrative example and highlights its benefits compared to baselines.

Our contributions are as follows: (i) We propose a new implicitly guided generative framework and provide general theoretical guarantees (Section 3.1). (ii) We show how to incorporate the proposed implicit guidance within diverse off-the-shelf generative modeling frameworks, e.g., variational autoencoders [21], flow matching [22], and walk-jump sampling [23] (Section 3.2). (iii) We show empirically that GPE models achieve competitive results in an *in silico* benchmark for offline (non-sequential) protein fitness optimization in two well-studied protein domains (Section 4.1). (iv) We show how GPE can "self train", pushing the boundaries of the designs' property beyond their scope in the training set (Section 4.2).

## 2  Related work

Most related works have been applied in the context of protein design and optimization. This problem has been tackled with a plethora of approaches, including latent space optimization [24, 11, 25, 15], reinforcement learning [6, 26, 15], genetic algorithm-based approaches [27, 8, 28, 29], energy-based methods [30, 31, 14], methods based on machine translation [32, 33, 12] and predictor-guided generative models [34, 7, 13, 35–39]. These approaches are typically deployed in either an offline setting, where optimization is performed on a fixed dataset, or an online setting involving iterative, multi-round optimization. While our work focuses on the former, we note that our model can be readily integrated into online frameworks, for instance, to warm-start the optimization or as a generative model to produce design candidates at each iteration.

---

[2]These discriminative models are called by different names, such as, predictor, scorer, surrogate, proxy, (pseudo-)oracle, etc.

Our work is closer to iterative editing methods, where the goal is to iteratively make local modifications to inputs for extrapolating the property of interest. For instance, Damani et al. [33] propose to continuously revising combinatorial structures on small molecules via paired data. Other works [12, 18] match sequences by generating perturbed sequences, scoring them with a (learned) scorer, and matching the original sequence with the generated ones (if the predicted score is below a certain threshold). Kirjner et al. [14] propose an approach where they iteratively use Gibbs sampling [40], followed by a scorer-based selection, on smoothed protein fitness landscapes. In contrast to previous work, GPE introduces a domain and modality-agnostic framework, comes with new theoretical guarantees that link generated samples to the property improvement. Moreover, unlike previous work, our method does not rely on discriminative models on any step of training, sampling or selection. Therefore, it can easily be integrated with previous work, potentially improving performance even further.

Our pairing step induces a preference relation, connecting our approach to preference learning [41], including DPO [42] for large language models, which shifts models toward preferred outcomes under (implicit) KL regularization. Recent protein-design work echoes this idea. Lee et al. [43] propose a fine-tuning framework to align protein language models to a specific desired fitness by ranking mutants. ReFT [44] filters data with auxiliary rewards and fine-tunes on the preferred subset for backbone generation. Zhou et al. [45] learns residue-level energy preferences to train a conditional diffusion model with a direct preference objective for antibody co-design and Mistani et al. [46] uses DPO on curated chosen–rejected receptor–binder pairs for peptide/protein binders. Despite these links, our setting differs from classical preference learning which compares pairs without enforcing proximity in data space or property magnitude.

Finally, we note that our contribution is orthogonal to classifier guidance [47] and classifier-free guidance [48] approaches, commonly used to guide diffusion models [49, 50] (and some instantiations of flow matching). In fact, our framework could seamlessly be integrated into both approaches, either by leveraging a trained classifier (classifier guidance) or by using the low-property seed as the conditioning signal (classifier-free guidance).

## 3 Implicit guided generation by data matching

### 3.1 Problem setting

We address the problem of *offline optimization* [51, 52], also known as model-based optimization, where the goal is to optimize a black-box function using only a fixed, static dataset. This paradigm stands in contrast to online optimization approaches, such as active learning or Bayesian optimization, which iteratively query a ground-truth objective function to acquire new data and refine a surrogate model.

Our goal is to generate new samples, referred to as *designs*, given one or more lead data points, referred to as *seeds*. The key objectives are twofold: a generated design must improve upon its seed in a specific property of interest (e.g., the potency of a molecule) while also adhering to predefined constraints (e.g., a limited number of modifications).

More formally, let $\mathcal{X}$ be a design space representing either a $d$-dimensional continuous space, $\mathbb{R}^d$, or a discrete space of length $L$ over a finite vocabulary $\mathcal{V}$ (e.g. for proteins $|\mathcal{V}| = 20$). Let $g : \mathcal{X} \to \mathbb{R}$ be a function that quantifies the property of interest. Given a seed $x \in \mathcal{X}$, our objective is to generate new candidate designs $x' \in \mathcal{X}$ satisfying two conditions: (i) the designs' score must exceed that of the seed, $g(x') > g(x)$, and (ii) the designs must satisfy a set of constraints, represented by a boolean function $C(x,x')$. For example, this function could encode a bound on the distance between seed and design, $C(x,x') = \text{dist}(x,x') < \Delta_x$ for some threshold $\Delta_x \in \mathbb{R}^+$.

In this work, we cast the problem above as a conditional generation problem, i.e., learn how to sample from the *improved distribution* given an initial seed $x$:

$$p^+(x'|x) = \frac{p(x')\mathbb{I}(x,x')}{Z(x)}, \quad Z(x) = \int p(x')\mathbb{I}(x,x')dx', \tag{1}$$

where the indicator $\mathbb{I}(x,x')$ is given by:

$$\mathbb{I}(x,x') = \begin{cases} 1, & g(x') > g(x) \land C(x,x'), \\ 0, & \text{otherwise.} \end{cases}$$

The normalizing factor $Z(x)$ in (1) is generally intractable due to the (usually) high dimensionality of the data and it cannot be solved directly. The following theorem establishes a learning criterion for the improved distribution.

**Theorem 1** (Optimality of implicit property enhancement). *Let $p(x)$ be a probability distribution and $p^+(x'|x)$ its corresponding improved distribution, as defined in (1). Consider the objective functional:*

$$\mathcal{L}(q) = -\mathbb{E}_{x \sim p(x)}\big[\mathbb{E}_{x' \sim p^+(x'|x)}[\log q(x'|x)]\big],$$

*where $q(\cdot\,|\,x)$ is any family of conditional densities with $\operatorname{supp}(p^+) \subset \operatorname{supp}(q)$. Then, the objective $\mathcal{L}(q)$ has the unique minimizer:*

$$q^*(x'\,|\,x) = p^+(x'|x).$$

*Proof.* Introduce a Lagrange multiplier $\lambda(x)$ for the constraint $\int q(x'\,|\,x)dx' = 1$. The Lagrangian is defined as:

$$J(q,\lambda) = -\int p(x)\int p^+(x'|x)\log q(x'\,|\,x)dx'dx + \int p(x)\lambda(x)\Big(\int q(x'\,|\,x)dx' - 1\Big)dx.$$

Taking the functional derivative w.r.t. $q(x'\,|\,x)$ and setting it to zero gives:

$$-\frac{p(x)p^+(x'\,|\,x)}{q(x'\,|\,x)} + p(x)\lambda(x) = 0 \quad\Longrightarrow\quad q(x'\,|\,x) = \frac{p^+(x'\,|\,x)}{\lambda(x)}.$$

Enforcing normalization $\int q(x'\,|\,x)\,dx' = 1$ forces $\lambda(x) = 1$. Hence $q^*(x'\,|\,x) = p^+(x'|x)$, as claimed. $\square$

We stress that the condition that the support of the variational density is greater than the support of the ground truth density is fairly standard in variational inference and can be easily ensured with appropriate architecture choices. Furthermore, we note that the proof is an adaptation of a similar approach for showing that maximum likelihood recovers the ground truth density at optimality.

**A one-dimensional example.** To build intuition, consider the following one-dimensional example: The data distribution $p(x)$ is the standard normal and the property of interest is defined as $g(x) = -(x-1)^2$. For a given suboptimal sample $y$, the improved distribution is defined as $p^+(x'\,|\,y) \propto p(x')\mathbb{I}\big(g(x') > g(y)\big)$. We now try to compute the conditional expectation $f^*(y) \approx \mathbb{E}\big[x'\,|\,g(x') > g(y)\big]$, the probability $p\big(g(x') > g(y)\big)$, and comparison of the quality $g(y)$ to the average quality of improved samples. From Figure 2, we observe a few things: (i) Conditional Expectation: When $y$ is far from the optimal region ($x \approx 1$), the expected superior $x'$ is significantly higher. As $y$ nears the optimum, the improvement diminishes. (ii) Probability of Improvement: The likelihood of finding a superior sample decreases as $y$ approaches the optimal quality and (iii) Quality Comparison: The plot shows that the improved samples not only exhibit higher $x'$ values but also consistently have enhanced quality $g(x')$ relative to $g(y)$.

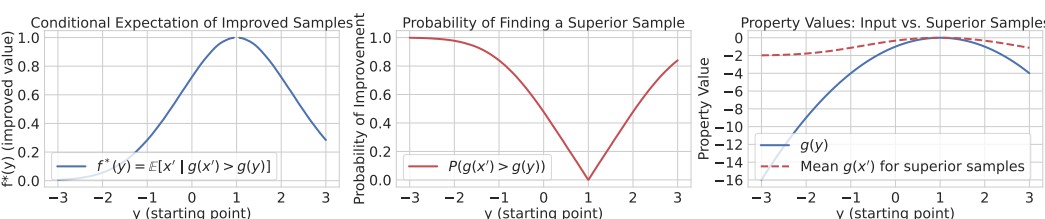

Figure 2: **Left:** Conditional expectation $f^*(y) = \mathbb{E}[x'\,|\,g(x') > g(y)]$ as a function of the starting sample $y$. This panel illustrates that when $y$ is far from the ideal (i.e., lower quality), the expected superior sample $x'$ is significantly higher, indicating a large potential for improvement. **Center:** Probability $p\big(g(x') > g(y)\big)$ of finding a superior sample $x'$ increases for lower-quality $y$ and decreases as $y$ approaches the optimal quality. **Right:** Comparison of the original property $g(y)$ with the mean property value of the superior samples $\mathbb{E}[g(x')\,|\,g(x') > g(y)]$ reveals that the improved samples indeed possess higher quality.

## 3.2 Matched generative models

Given an initial dataset $\mathcal{D} = \{(x_i, g(x_i))\}_{i=1}^n$, $x_i \in \mathcal{X}$, $g(x_i) \in \mathbb{R}$ and a set of constraints $C(x, x') : \mathcal{X} \times \mathcal{X} \to \{\text{True}, \text{False}\}$, we construct a *matched dataset* $\mathcal{M} = \{(x_i, x_i')\}_{i=1}^N$, following Tagasovska et al. [19]:

$$\mathcal{M} = \{(x, x') \in \mathcal{X} \times \mathcal{X} : g(x') > g(x) \wedge C(x, x')\}, \tag{2}$$

where $x$ and $x'$ are elements of $\mathcal{D}$. By construction, $\mathcal{M}$ contains samples drawn from $p^+(x' \mid x)p(x)$. We stress that typically the matched dataset $\mathcal{M}$ has the attractive property that it is significantly larger than the initial dataset $\mathcal{D}$, i.e. $N > n$, which is a particularly desirable in the low-data regime.

Equipped with Theorem 1 and matched datasets (2), our goal is to learn a conditional generator $q_\theta(x' \mid x)$, parameterized by $\theta$, that approximates the improved distribution $p^+(x' | x)$. This is done by minimizing the following loss:

$$\mathcal{L}(\theta, \mathcal{M}) = - \sum_{(x, x') \in \mathcal{M}} \log q_\theta(x' \mid x). \tag{3}$$

The minimization of $\mathcal{L}(\theta, \mathcal{M})$ aligns the generative process with the desired improvement through conditional density estimation, bypassing the need for an explicit property predictor. In some cases (e.g. discrete data), one can directly parametrize the likelihood above. When that is not easily feasible, we resort to approximating it. Next, we show how we can adapt off-the-shelf models to sample from the conditional distribution of interest. We note that our approach is fairly general and can similarly be applied to other generative models, such as autoregressive or diffusion models.

**Matched PropEn (mPropEn).** PropEn [19] leverages matched datasets to approximate the gradient of the property of interest by minimizing a matched reconstruction loss, between a sample with low property and its higher-value match. The trained model is used to sample designs by "following the gradient": given an initial seed, the model is applied iteratively, generating one sample per iteration step. When data is discrete, the matched reconstruction loss *is* the negative log-likelihood loss in (3). Here we extend the gradient-based PropEn into an instantiation of matched generative models[3]: we sample set of designs from the logits of the model (by choosing an appropriate temperature).

**Matched variational auto-encoder (mVAE).** We instantiate $q_\theta(x' \mid x)$ as a latent-variable model defined as:

$$q_\theta(x' \mid x) = \int p_\psi(x' \mid x, z) q_\phi(z \mid x, x') dz,$$

parameterized by $\theta = (\phi, \psi)$, where $\phi$ and $\psi$ are parameters of the encoder and decoder, respectively. Training is done by minimizing the following negative ELBO [21, 53]:

$$\mathcal{L}_{\text{mVAE}}(\theta, \mathcal{M}) = - \sum_{(x, x') \in \mathcal{M}} \mathbb{E}_{q_\phi(z \mid x, x')} \log p_\psi(x' \mid x, z) + \text{KL}\big(q_\phi(z \mid x, x') \,\|\, \mathcal{N}(z; 0, I_d)\big), \tag{4}$$

where $\mathcal{N}(z; 0, I_d)$ is the standard $d$-dimensional normal distribution. This objective is a lower bound of the maximum-likelihood problem in Theorem 1. Hence, the mVAE approximates the optimal density while providing a tractable latent representation and sampling mechanism. We sample designs following the standard (conditional) VAE approach [21, 54]: given a query seed $x$, sample $z$ from the prior, then forward through the decoder to get approximated samples from $p_\psi(x' | x, z)$.

**Matched flow matching (mFM).** We use a continuous normalizing flow [55] to generate samples from the improved conditional distribution by, given an initial seed, (i) first sample from an (easy) source distribution $p_0$, (ii) then transform it into a sample from the desired target distribution $p^+(x' | x)$ by solving an ODE.

Until recently, continuous normalizing flows have been trained by directly optimizing the maximum likelihood objective of Theorem 1. However, recently practitioners have switched to the flow matching objective as it allows simulation-free training and tends to produce higher likelihood in practice [22, 56, 57]. In flow matching, one uses a time-dependent velocity field $v_t^\theta(x_t', x)$, with $t \in [0, 1]$, and $x_t'$ an intermediate sample from the probability path $p_t(x_t' | x)$ that defines the velocity

---

[3]We abuse notation by calling this model "matched PropEn", since [19] also leverages matched datasets. The difference is how we use the model to sample designs.

field. We follow [22] and choose the linear probability path. The velocity field is learned by adapting the conditional flow matching loss [22]—that matches the parameterized velocity to the optimal velocity that transports $p_0$ to $p^+(\cdot|x)$—to our setting:

$$\mathcal{L}_{\mathrm{mFM}}(\theta,\mathcal{M}) = \sum_{(x,x')\in\mathcal{M}} \mathbb{E}_{t\sim U(0,1),x_0'\sim p_0}||v_t^\theta(x_t',x)-(x'-x_0')||^2, \tag{5}$$

where $x_t' = (1-t)x_0' + tx'$. In case data is discrete, we can equivalently use the discrete flow matching loss [58, 59]. We sample improved designs by solving the ODE $\dot{x}_t' = v_t^\theta(x_t',x)$, where $x_0' \sim p_0$ and $x_1'$ is an (approximate) sample from $p^+(x'|x)$.

**Matched walk-jump sampling (mWJS).** Walk-jump sampling [23] is a score-based generative model that approximate samples from $p^+(x'|x)$ by following a two-step procedure, given a (usually high) noise level $\sigma$: (i) sample from the smooth distribution $p(y|x) = p^+(x'|x)\star\mathcal{N}(0,\sigma^2 I_d)$, (ii) then estimate samples $\hat{x} = \mathbb{E}(x|y)$. We approximate the score function with conditional denoiser $D_\theta : \mathbb{R}^d \times \mathbb{R}^d \to \mathbb{R}^d$, a neural network parameterized by $\theta$, that takes as input pairs $(y,x)$ and outputs an estimated clean version of $x'$ (the clean version of the noisy design). The denoiser is trained by minimizing the following loss:

$$\mathcal{L}_{\mathrm{mWJS}}(\theta,\mathcal{M}) = \sum_{(x,x')\in\mathcal{M}} \mathbb{E}_{\varepsilon\sim\mathcal{N}(0,I_d)}||D_\theta(x'+\sigma\varepsilon,x)-x'||^2. \tag{6}$$

Following learning the conditional denoiser, we approximate the score function $p(y|x)$ using the conditional version of Tweedie-Miyasawa formula [60, Proposition 1], i.e.,

$$\nabla\log p(y|x) \approx s_\theta(y|x) := (D_\theta(y,x)-y)/\sigma^2.$$

We then leverage the (learned) conditional denoiser ($D_\theta$) and the score function ($s_\theta$) to generate designs ($x'$) conditioned on seeds ($x$):

  (i) *(init.)* pick an initial seed $x$ and initialize $y_0$ with noise.

  (ii) *(walk)* sample $y_k \sim p(y|x)$ with Langevin MCMC with discretization step $\delta$[4]:

$$y_{k+1} = y_k + \delta s_\theta(y|x) + \sqrt{2\delta}\varepsilon_k,\ \varepsilon_k \overset{\mathrm{iid}}{\sim} \mathcal{N}(0,I_d).$$

  (iii) *(jump)* generate clean designs at arbitrary step $K$: $x_K' \leftarrow D_\theta(y_K,x)$.

Note that the least-squares loss (6) does not correspond to maximum likelihood; however, samples from this model are approximate draws from $p^+(x'|x)$. The degree of this approximation is controlled by $\sigma$ [62]. In practice, $\sigma$ plays the role of a regularizer, both for learning a smoother density and for easing the problem of sampling. Due to the simplicity of this scheme ($\sigma$ is the main hyperparameter), it has proven to be an effective sampling strategy in practical applications [63, 30, 60, 64].

### 3.3 Iterative sampling

Our matched generative models sample designs, given seeds, with improved property and satisfying constraints $C(x,x')$. They are amenable to iterative optimization and, in practice, we apply the sampling procedure multiple times, (implicitly) guiding designs toward higher property values, until some criterion is reached (e.g., the number of iteration, a certain value on the property). Similar to [19], and unlike e.g. [12, 14], our iterative optimization approach does not rely on any auxiliary discriminative models. Unlike [19], the matched models generate a population of designs at each iteration. We propose a simple iterative process: $x_{k+1}' \sim p^+(\cdot|x_k')$, where $x_k'$ is an element of design set $\mathcal{D}_k$ and $k = 0,1,...,K$. Starting from a set of seeds $\mathcal{D}_0 = \{x_i\}_{i=1}^N$ (including the case of a single seed), we get a population of improved designs $\mathcal{D}_1$ after the first iteration, which are then used as seeds on the following iteration and the process is repeated. The samples on iteration $k+1$ should have a better property than the samples on iteration $k$, while still satisfying additional constraints. Algorithm 1 in appendix describes a simple pseudo-code for this procedure.

---

[4]The kinetic Langevin MCMC [61] has better mixing properties, which we do not discuss here due to space.

### 3.4 Iterative training on self-generated data

Starting from a small set of $N_0$ true matched data points $\mathcal{M}_0 = \{(x_0, x'_0)_i\}_{i=1}^{N_0}$, for every step $k$ the model generates new samples optimized for desired properties and pairs them with their original inputs to form *pseudo-matched* examples $\{(x_k, x'_k)_i\}_{i=1}^{N_k}$. These synthetic matches are then merged back into the training set, $\mathcal{M}_k \leftarrow \mathcal{M}_{k-1} \cup \{(x_k, x'_k)_i\}_{i=1}^{N_k}$, effectively enlarging the dataset. This iterative cycle—generate, match, retrain—enables the model to (i) refine its improved target distribution and (ii) explore regions

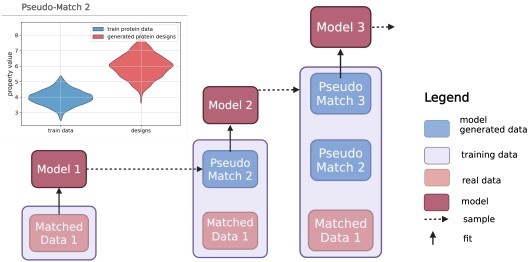

Figure 3: Iterative training with pseudo-matches.

of data space beyond the original manifold. As a result, generative performance steadily improves: the model learns from its own best guesses, sharpens its ability to produce high-value samples, and is able to extrapolate property values than could not have been achieved with the initial, limited ground-truth set alone. This procedure is summarized in Figure 3 and Algorithm 2.

Our approach draws from classic semi-supervised *pseudo-labeling* methods [65, 66] and their modern successors in generative contexts like FixMatch [67], where models reuse their own predictions for training models for supervised tasks. While recent studies highlight failure modes in high-capacity large language models—regression to overly common patterns and failure to explore richer outputs [68, 69]—our work contrasts by focusing on *sparse, low-data* settings (e.g. early-stage drug or antibody design). Here, we use iterative pseudo-matching to bootstrap new data pairs, mitigate regression to the mean, and push property values beyond the original training manifold, crucially operating without the oracles or pruning methods prevalent in many LLM-based model collapse solutions.

## 4 Experimental results

We evaluate the effectiveness of our models on two *in silico* offline (non-sequential) settings related to protein design. First, we compare our model examples with baselines on a protein fitness optimization benchmark, focusing on subdomains of AAV and GFP proteins (Section 4.1). Then, we show how iterative training on self-generated data can push the boundaries of the designs' properties beyond their scope in the training set (Section 4.2). We emphasize our approach is general and can be applied in modalities other than biological sequences, illustrated on a MNIST toy task presented in Appendix F.

### 4.1 Protein fitness optimization

**Datasets.** We evaluate the performance of our model on two important protein subdomain datasets: adeno-associated virus (*AAV*) [70] and green fluorescent protein (*GFP*) [71]. The former measures the ability of the AAV to package a DNA payload, for applications in e.g. gene delivery, while the latter's fitness is its fluorescence properties, useful for e.g. biomarkers. AAV and GFP datasets contain 44,156 and 56,806 pairs of amino acid sequences and experimentally measured property, respectively.

We use the benchmark proposed by Kirjner et al. [14], which contains functional segments of length $L = 28$ and $L = 237$ amino acids for AAV and GFP, respectively. Following common practice, the protein sequences are one-hot encoded over a dictionary of 20 amino acids. The optimal fitness set for each dataset is defined as the top 99th percentile of experimental data with the highest measured property. We consider the two splits proposed by the authors—"medium" and "hard" splits—with 2,139/3,448 samples for AAV and 2,828/2,426 for GFP. The medium split is a subset of the data containing the 20th-40th percentiles that are 6 edit distances or more from any sample in the optimal fitness set. The hard split contains the lowest 30th percentiles that are 7 mutations or more away from the optimal fitness set. Similar to [19], we consider the following constraints when creating the matched datasets for our models: $C(x, x') = \text{dist}(x, x') < \Delta_x \wedge (g(x') - g(x) < \Delta_g)$, where we consider Levenshtein distance, $\Delta_x$ of 5 and 10 and $\Delta_g$ of .2 and .05 (before normalization) for AAV and GFP tasks, respectively. These thresholds were selected based on an ablation conducted with one model variant, mWJS, on the medium splits (see Appendix C.2).

**Baselines.** We compare our GPE models proposed in the previous section with a variety of methods in the offline optimization setting[5]: GFlowNets (GFN-AL) [72], model-based adaptive sampling

---

[5]The baseline results are taken from [14].

(CbAS) [7], greedy search (AdaLead) [8], a simple Bayesian optimization baseline (BO-qei) [73] (using the implementation of [51]), conservative model-based optimization (CoMs) [9], proximal exploration (PEX) [28], Gibbs with gradients [40] (GWG) adaptation from [14] and its labeled-smoothed version (GGS) [14]. We emphasize that, unlike our models, all baselines rely on a discriminative model either as part of the sampling process or as a way to select designs at each iteration.

**Evaluation and metrics.** Following [72, 14], each method starts the optimization process with 128 seeds, $X = \{x_i\}$, and generates 128 designs, $X^{'} = \{x_i'\}$, ideally with higher fitness than the initial seeds. For our experiments, we used the top 128 seeds from each dataset, but as shown in the Appendix C.3, our method is robust and performs well regardless of the initial seed selection.

The fitness of generated designs are approximated with a fitness predictor, $\hat{g}$. We use the fitness predictor (and model weights) from [14]. This (pseudo-)oracle is a 1D convnet trained on all wet-lab data provided by [70, 71] and used only for evaluation[6]. The fitness is min-max normalized on all tasks. For fair comparison with previous work, we use the same metrics as in [14][7]. Given the set of generated designs, we compute: $Fitness = \text{median}(\{\hat{g}(x_i')|x' \in X^{'}\})$, the median of the approximated fitness predicted by the pseudo-oracle (higher is better), $Diversity = \text{median}(\{\text{dist}(x, \tilde{x}) \,|\, x, \tilde{x} \in X^{'}, x \neq \tilde{x}\})$, the median of the Levenshtein distance between every pair of sequence on the generated set, and $Novelty = \text{median}(\{\eta(x_i', X)\}_{i=1}^{128}))$, where $\eta(x, X) = \min(\{\text{dist}(x, \tilde{x}) \,|\, \tilde{x} \in X, x \neq \tilde{x}\})$ is the minimum distance of sample $x$ to any starting seed in $X$. We are interested in designs that have higher fitness, while diversity and novelty are shown to assess exploration/exploitation tradeoffs. As pointed by [14], random sequences would provide very high diversity and novelty and unreliable (predicted) fitness.

**Implementation details.** For the iterative sampling, we start with 128 seeds from the training set (similar to other baselines) and sample designs according to Algorithm 1 for $K = 20$ iterations. At each iteration, we sample a pool of $M = 2560$ designs and reject the repeated ones and those that have a Levenshtein distance larger than 10 from any seed. On the final iteration, we randomly pick 128 samples from the last pool of designs. For details on the architecture, training and sampling hyperparameters of our models, see Appendix B.3.

**Results.** Table 1 and Table 2 compare our matched generative models with baselines on both splits of AAV and GFP datasets, respectively. Our matched generative models achieve similar or better results than baselines on most tasks, while being conceptually simpler and without relying on any external predictor. We remark that the main contribution of the best performing baseline (GSS), i.e. graph-based smoothing of the fitness landscape, is orthogonal to our approach and could potentially be integrated to our matched models to further improve results.

Figure 4 compares one of our proposed models, mWJS, to its unmatched counterparts, akin to a simplified version of [30]. The two models have the same architecture and hyperparameters, the only difference being that the former is conditioned on the matched dataset while the latter is an unconditional model. It clearly illustrates the benefits of our approach: the median fitness of designs generated by the matched model increases as we do more sampling iterations (until it plateaus), while the fitness of the unmatched one remains mostly unchanged. Table 3 and Table 4 (appendix) shows how the matched version of the generative models improve over the unconditional version for the generative models considered. Finally, Figure 8 (appendix) shows quantitative examples of how the distribution changes over iterations starting from randomly chosen single seeds (not seen during training).

## 4.2 Iterative training on self-generated data

**Dataset.** In this task, we focus on optimizing therapeutic proteins—specifically, antibodies—using the mutagenesis library introduced in [74]. This dataset covers three different antigen targets; here, we concentrate on the HER2 subset[8]. Physicochemical properties such as hydrophobicity and electrostatic charge strongly influence a molecule's developability profile, impacting key attributes like viscosity and specificity [75]. Unlike binding affinity or expression, which lack reliable computational proxies, these physicochemical features can be estimated *in silico* via closed-form metrics that correlate well with experimental measurements. For clarity, we use charge at given pH of a protein to illustrate the

---

[6]Using trained models to approximate fitness is obviously unreliable and should be taken with a grain of salt. It does provide, however, a convenient way to compare different methods in a toy-task setting.

[7]We use the implementation provided by the authors in https://github.com/kirjner/GGS.

[8]Human epidermal growth factor receptor 2, oncogene with an important role in the development and progression of certain aggressive types of breast cancer.

| | Medium difficulty | | | Hard difficulty | | |
|---|---|---|---|---|---|---|
| Method | Fitness | Diversity | Novelty | Fitness | Diversity | Novelty |
| GFN-AL [72] | 0.20 (0.1) | 9.6 (1.2) | 19.4 (1.1) | 0.10 (0.1) | 11.6 (1.4) | 19.6 (1.1) |
| CbAS [7] | 0.43 (0.0) | 12.7 (0.7) | 7.2 (0.4) | 0.36 (0.0) | 14.4 (0.7) | 8.6 (0.5) |
| AdaLead [8] | 0.46 (0.0) | 8.5 (0.8) | 2.8 (0.4) | 0.40 (0.0) | 8.5 (0.1) | 3.4 (0.5) |
| BOqei [73] | 0.38 (0.0) | 15.2 (0.8) | 0.0 (0.0) | 0.32 (0.0) | 17.9 (0.3) | 0.0 (0.0) |
| CoMS [9] | 0.37 (0.1) | 10.1 (5.9) | 8.2 (3.5) | 0.26 (0.0) | 10.7 (3.5) | 10.0 (2.8) |
| PEX [28] | 0.40 (0.0) | 2.8 (0.0) | 1.4 (0.2) | 0.30 (0.0) | 2.8 (0.0) | 1.3 (0.3) |
| GWG [40] | 0.43 (0.1) | 6.6 (6.3) | 7.7 (0.8) | 0.33 (0.0) | 12.0 (0.4) | 12.2 (0.4) |
| GGS [14] | 0.51 (0.0) | 4.0 (0.2) | 5.4 (0.5) | **0.60** (0.0) | 4.5 (0.5) | 7.0 (0.0) |
| **GPE (ours)** | | | | | | |
| **mPropEn** | 0.52 (0.02) | 6.4 (0.7) | 6.0 (0.7) | 0.38 (0.04) | 8.7 (0.7) | 7.8 (0.7) |
| **mVAE** | 0.48 (0.02) | 9.5 (0.3) | 6.0 (0.0) | 0.38 (0.04) | 12.0 (1.2) | 7.3 (0.8) |
| **mFM** | 0.52 (0.01) | 6.2 (0.2) | 5.6 (0.6) | 0.35 (0.02) | 6.6 (0.3) | 5.2 (0.5) |
| **mWJS** | **0.53** (0.01) | 5.2 (0.2) | 5.6 (0.6) | 0.54 (0.04) | 4.6 (0.7) | 6.6 (0.5) |

Table 1: AAV benchmarks results. Results are shown with mean/standard deviation across 5 runs. We **bold** and underline the best and second best fitness score, respectively. Unlike our models (bottom), all baselines use discriminative model to guide sampling. Details about baselines and metrics can be found in Section 4.1.

| | Medium difficulty | | | Hard difficulty | | |
|---|---|---|---|---|---|---|
| Method | Fitness | Diversity | Novelty | Fitness | Diversity | Novelty |
| GFN-AL [72] | 0.09 (0.1) | 25.1 (0.5) | 213 (2.2) | 0.10 (0.2) | 23.6 (1.0) | 214 (4.2) |
| CbAS [7] | 0.14 (0.0) | 9.7 (1.1) | 7.2 (0.4) | 0.18 (0.0) | 9.6 (1.3) | 7.8 (0.4) |
| AdaLead [8] | 0.56 (0.0) | 3.5 (0.1) | 2.0 (0.0) | 0.18 (0.0) | 5.6 (0.5) | 2.8 (0.4) |
| BOqei [73] | 0.20 (0.0) | 19.3 (0.0) | 0.0 (0.0) | 0.00 (0.5) | 94.6 (71) | 54.1 (81) |
| CoMS [9] | 0.00 (0.1) | 133 (25) | 192 (12) | 0.00 (0.1) | 144 (7.5) | 201 (3.0) |
| PEX [28] | 0.47 (0.0) | 3.0 (0.0) | 1.4 (0.2) | 0.00 (0.0) | 3.0 (0.0 | 1.3 (0.3) |
| GWG [40] | 0.10 (0.0) | 33.0 (0.8) | 12.8 (0.4) | 0.00 (0.0) | 4.2 (7.0) | 7.6 (1.1) |
| GGS [14] | 0.76 (0.0) | 3.7 (0.2) | 5.0 (0.0) | 0.74 (0.0) | 3.6 (0.1) | 8.0 (0.0) |
| **GPE (ours)** | | | | | | |
| **mPropEn** | 0.62 (0.02) | 4.2 (0.3) | 8.0 (0.4) | **0.88** (0.02) | 2.8 (0.2) | 7.00 (0.0) |
| **mVAE** | **0.84** (0.03) | 1.9 (0.2) | 7.0 (0.7) | 0.78 (0.04) | 1.3 (0.2) | 7.5 (0.6) |
| **mFM** | 0.50 (0.03) | 5.3 (0.2) | 7.0 (0.0) | 0.55 (0.04) | 5.4 (0.1) | 7.7 (0.5) |
| **mWJS** | 0.76 (0.03) | 3.2 (0.1) | 6.0 (0.0) | 0.78 (0.02) | 2.9 (0.2) | 7.0 (0.0) |

Table 2: GFP benchmarks results. Results are shown with mean/standard deviation across 5 runs. We **bold** and underline the best and second best fitness score, respectively. Unlike our models (bottom), all baselines use discriminative model to guide sampling. Details about baselines and metrics can be found in Section 4.1.

benefits of self-training with exact evaluation. Each antibody is represented in AHo numbering format [76], yielding one-hot encoded sequences of length 298. From the original training set, we randomly select 1,000 examples to mirror the low-data regimes common in real-world drug discovery—often on the order of only a few hundred samples. We use $\Delta_x = 10$ and $\Delta_g = 0.05$ for matching constraints.

**Metrics.** We evaluate the performance in terms of per-design metrics: *average improvement* (AI), the difference in property value between each design and its corresponding seed, and *ratio of improvement* (RI), the proportion of designs which have improved property values compared to their seeds.

**Implementation details.** Our architecture and training procedures follow Tagasovska et al. [19]. The overall pipeline is illustrated in Figure 3. At round $k = 1$, we train GPE models on the dataset of true matched points, generated from $N_0 = 400$ sequences, and evaluate on a held-out test set of 100 examples. Then, for each round $k$ (for a total of $K = 4$ rounds), we sample new designs, generate new pseudo-match pairs, update the training set and retrain the models.

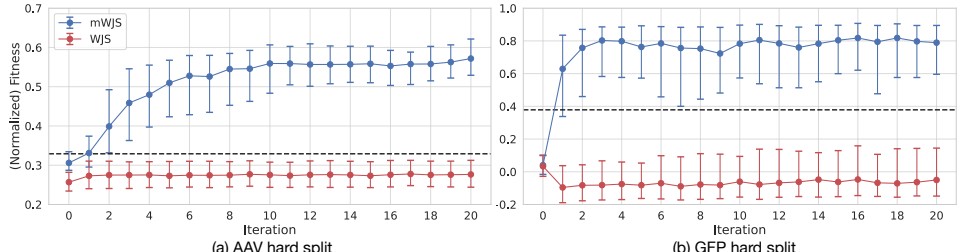

Figure 4: Normalized fitness per iteration (for a total of 20 iterations) for AAV (a) and GFP (b) hard splits. We show results for WJS (red) and mWJS (blue). The dashed lines correspond to the fitness of training set sample with maximum fitness. For each iteration, the median and 25-75 percentiles are shown. We start with 128 seeds from training set (iteration 0) and iteratively sample designs for a total of 20 iterations.

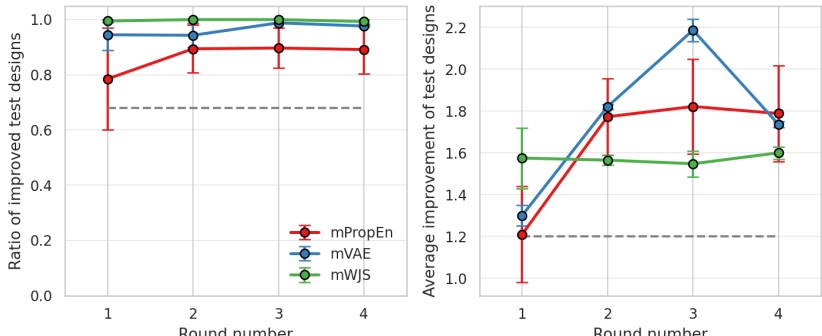

Figure 5: Iterative training with three GPE models starting from only 400 examples. Each panel shows how RI (left) and AI (right) metrics evolve over four rounds. The dashed lines show performance of a model trained on 1,000 samples. All methods rapidly surpass this baseline by round 2 and continue to improve until saturating around round 4.

**Results.** Figure 5 demonstrates that across three GPE variants, when starting from only 400 seed sequences, each successive self-training iteration outperforms a baseline trained on 1,000 true examples, both in the fraction of improved designs and in average property gain. Moreover, with each round, the property ceiling rises until it plateaus around round 4, consistent with our limited initial dataset. These results underscore the power of iterative self-training in low-data regimes typical of drug discovery. Appendix D.1 shows an ablation comparing our method with a version where designs for each round are selected using a (trained) predictor or the ground-truth oracle.

## 5    Conclusion

We introduced generative property enhancer (GPE), a novel approach for guiding generative models that circumvents the need for explicit discriminative models, which are often unreliable in low-data regimes and for tail-end distribution properties. GPE offers a simple, scalable, end-to-end, and broadly adaptable solution compatible with diverse generative models and data representations. GPE achieves competitive performance in a standard offline (non-sequential) protein fitness optimization benchmark and can be iteratively trained on self-generated data, successfully applied to an antibody property optimization task.

Our work shares similar limitations to related approaches. First, GPE currently focuses on single-property enhancement, which may not fully meet industry expectations for designs that must simultaneously adhere to multiple properties of interest. Second, our current validation has thus far been conducted on *in silico* benchmarks. Finally, the iterative sampling process involves some hyperparameter tuning (e.g., number of iteration steps, number of seeds per step). We observed, however, that these hyperparameters are transferable between the protein tasks studied, suggesting some degree of robustness. Future work will focus on extending our framework to multiple-property optimization, leveraging domain-specific knowledge for targeted applications, and experimenting with multi-modal representations.

## Broader Impact

Design optimization is an important problem in many scientific applications, such as protein engineering, synthetic biology, materials, environment and molecule design. These are very long and challenging endeavors that involve many steps to achieve success. In this paper we propose an approach to this problem, which deals with one of these steps. There is still a lot of work that needs to be done to validate these kinds of models in practice (e.g., lab experimental validation, clinical trials, technological advances, etc). That being said, if successful, advances in this field can directly impact quality of human life. Like many other powerful technologies, we need to ensure that these models are deployed in ways that are safe, ethical, accountable and exclusively beneficial to society.

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

# Appendix

## A  2D toy task

Here, we analyze the behavior of one of our models—the matched walk-jump sampling—on a simple task in 2 dimensions. We start with a dataset of 186 points whose property value increases as the points move counter clockwise on the spiral. We then create the matched dataset considering a threshold $\Delta_x = 0.5$, $\Delta_y = 0.01$ and the Euclidean distance between points, with a total of 498 pairs (we use 401 for training). Figure 6 shows the initial and the matched dataset. We proceed by training a mWJS (we use a simple 2 layers MLP as the denoiser) on the matched dataset and use this model to sample new points with (hopefully) increased properties.

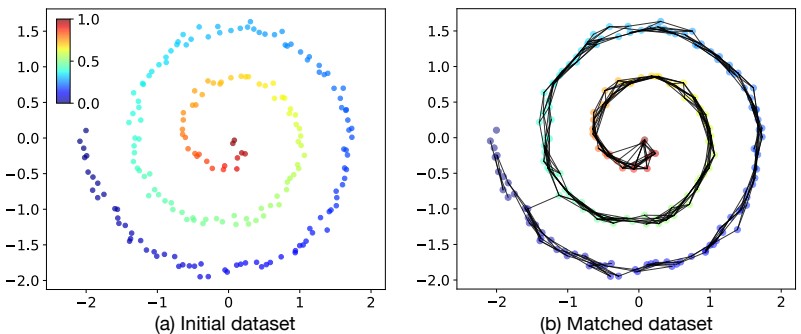

Figure 6: (a) Initial dataset, 186 points. (b) Matched dataset, 498 pairs. The data coordinates are between -2 and 2 and the property values range from 0 to 1, increasing counter-clockwise.

Figure 7 shows examples of generated designs following our iterative sampling over 50 optimization iterations, starting from unseen test points. We observe exactly what we would expect: (i) the generated samples remain on the manifold of data, (ii) the property of the samples increase as we sample for more iterations, and (iii) samples at each iteration are diverse.

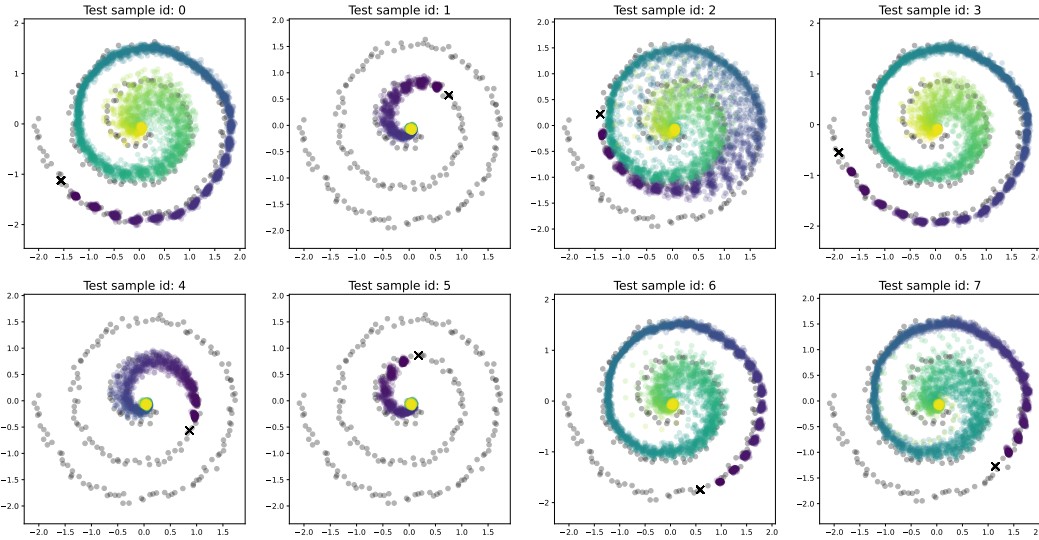

Figure 7: Examples of generated samples with our model for different initial seeds (marked as a cross on the plots). Samples from each iteration are color-coded (blue for first iteration, yellow for the last). The generated samples (i) remain on the manifold of data, (ii) are diverse, and (iii) increase property as we increase the number of iterations.

# B  Implementation details

## B.1  Iterative sampling algorithm

See Algorithm 1.

---
**Algorithm 1:** Iterative Sampling

---
**Data:** Model $q_\theta$, seeds $\mathcal{S} = \{x_i\}_{i=1}^n$, n iterations $K$, n samples per iteration $M$, distance threshold $\delta$, n designs $N$.
**Result:** Designs $\{x_i'\}_{i=1}^N$.

1   $\mathcal{D}_0 \leftarrow \mathcal{S}$;
2   **for** $k \leftarrow 0$ **to** $K-1$ **do**
     // 1.  Sample M designs given $\mathcal{D}_k$
3     $\mathcal{D}_{k+1} = \{x_i'\}_{i=1}^M \sim q_\theta(\cdot \,|\, \mathcal{D}_k)$;
     // 2.  rejection step
4     $\mathcal{D}_{k+1} \leftarrow \mathrm{unique}(\mathcal{D}_{k+1})$;
5     $\mathcal{D}_{k+1} \leftarrow \{x \,|\, \mathrm{dist}(x, \tilde{x}) < \delta, x \in \mathcal{D}_{k+1}, \tilde{x} \in \mathcal{S}\}$;
6   **end for**
7   $\mathcal{D}_K \leftarrow \mathrm{choose\_random}(\mathcal{D}_K, N)$ // N random samples from pool of designs
8   **return** $\mathcal{D}_K$

---

## B.2  Iterative training on self-generated data

See Algorithm 2.

---
**Algorithm 2:** Iterative training on self-generated data

---
**Data:** Initial parameters $\theta^{(0)}$; data distribution $p(x)$; conditional generator $q_\theta(x' \,|\, x)$; neighborhood radius $\Delta_x$; iterations $N_{\mathrm{iter}}$; samples per round $M$; loss $\mathcal{L}(\theta; \cdot)$; initial matched dataset $\mathcal{M}^{(0)}$.
**Result:** Trained parameters $\theta^{(N_{\mathrm{iter}})}$.

1   **for** $k \leftarrow 0$ **to** $N_{\mathrm{iter}} - 1$ **do**
     // 1.  Generate candidate samples
2     Draw $\{x_i\}_{i=1}^M \overset{\mathrm{i.i.d.}}{\sim} p(x)$;
3     For each $i$, draw $x_i' \sim q_{\theta^{(k)}}(x' \,|\, x_i)$;
     // 2.  Filter / enforce neighborhood (and optional property)
4     $\mathcal{D}^{(k)} \leftarrow \{(x_i, x_i') : d(x_i, x_i') \leq \Delta_x \wedge g(x_i') > g(x_i)\}$;
     // (Optional) discriminator/oracle filter:  keep only $(x_i, x_i') \in \mathcal{D}^{(k)}$ with score $s(x_i, x_i') \geq \tau$
     // 3.  Augment training set
5     $\mathcal{M}^{(k+1)} \leftarrow \mathcal{M}^{(k)} \cup \mathcal{D}^{(k)}$;
     // (Optional) deduplicate near-duplicates in $\mathcal{M}^{(k+1)}$
     // 4.  Retrain on full set (real + pseudo-matches)
6     $\theta^{(k+1)} \leftarrow \underset{\theta}{\arg\min} \, \mathcal{L}\left(\theta; \mathcal{M}^{(k+1)}\right)$   (initialized at $\theta^{(k)}$);
7   **end for**
8   **return** $\theta^{(N_{\mathrm{iter}})}$

---

## B.3  Matched generative models implementation details

The models on this paper were trained using single A100 Nvidia GPUs and 4 CPU workers per model. Training of different models varies, but they all were trained on less than 1 day. The hyperparameters for training and sampling are the same for all tasks, unless stated differently. They were chosen using a subset of the AAV hard task.

**mVAE.** We use the ResNet architecture proposed in [19], which has shown to be an efficient encoder-decoder framework for protein design. We extend this to a matched version by implementing a conditional VAE, with the conditioning on $x$ being concatenated to $z$ the latent representation. We have one layer of one-hot encoding, followed by 3-layer MLP resnet blocks with internal layers of size 128. Each mVAE was trained with Adam, learning rate $1e^{-4}$ for AAV and antibodies, and, $1e^{-5}$ for GFP, and train for 500 epochs each.

**mWJS.** Similar to [30], we model amino acid sequences (both seeds and designs) as one-hot encoding lying on the continuous space of dimension $d = 20 \times L$. We chose a noise level $\sigma = .5$ and generate noisy samples by adding gaussian noise to designs, i.e., $y = x' + \sigma\varepsilon, \varepsilon \sim \mathcal{N}(0, I_d)$. We adapt the architecture of the condition denoiser used for conditional walk-jump sampling from [60] to our setting, i.e., we modify their architecture to be applied to 1D sequences instead of 3D voxel grids. First we pad the two inputs (noisy design $y$ and clean seed $x$) so that they have lengths divisible by 8 (required for the denoiser) so that $L$=32 for AAV dataset, and 240 for GFP. The padded inputs are then forwarded through a single 1D conv (with kernel size 1) and added together (or we just encode $y$ in the unconditional setting). The resulting embedding is then forwarded through a 1D U-Net. Here, we adapt the 2D unet proposed by [77] to our 1D setting[9]. We remove the noise-conditioning, as our denoiser is trained on a single noise level. The architecture starts with 32 and contains 4 resolution levels, each level containing 4 residual blocks. At each resolution level, we downsample (upsample in the case of decoder) the sequence embedding by two while doubling (halving for the decoder) the number of channels. The two last layers of the encoder (two first layers of the decoder) contain self-attention blocks, similar to [77]. The denoiser model has a total of 3.8M parameters, and it is trained with batch size of 256, learning rate $1e^{-3}$, Adam [78] optimizer and a total of 5,000/1,000 epochs for AAV and GFP, respectively.

Sampling is done following the conditional walk-jump sampling from [60] (see Algorithm 1 in the paper). We use underdamped Langevin MCMC with the discretization scheme proposed by [61] with step size $\delta = \sigma/2$ and set $\gamma = 1/\delta$. Each iteration consists of two walk steps followed by a jump step, which approximates improved designs. The chains in the first iteration are initialized with a seed from the initial set of seeds and uniform + gaussian noise, $(x, y_0)$, where $y_0 = \mathcal{U}_{I_d}(0,1) + \mathcal{N}(0, \sigma^2 I_d)$. Each iteration consists of 2 walk steps followed by a jump step, which generates the new sets of design for iteration $i$. We use the designs generated at iteration $i$ to condition the chains on iteration $i+1$, until we reach 20 iterations. Note that we do not re-initialize the noisy seeds $y$ at each iteration: they keep being updated as the MCMC chain progresses.

**mFM.** Similar as above, we use the same data representation and the unet architecture of [77] adapted to the 1D setting. The conditioning mechanism is identical to the original implementation: at each scale we integrate embeddings for the time $t$ and seed $x$ (instead of the class label, as in the original architecture). As we represent molecules as discrete data, we use the discrete flow matching [58, 59] variant. In preliminary experiments, we observed the uniform source distribution $p_0$ achieved better performance than the masked distribution (where all tokens are masked). We use the implementation from [57][10], and in particular, their proposed `MixtureDiscreteProbPath` path with `PolynomialConvexScheduler` scheduler ($n = 2$) and the `MixturePathGeneralizedKL` loss. We also use their provided ODE solver with 32 steps (`ConditionalMixtureDiscreteEulerSolver`). Similar to the mWJS variant, we train for 5,000/1,000 epochs for AAV and GFP, respectively, we use the learning rate of $1e^{-3}$, Adam optimizer and batch size 256.

## C    Additional results on protein fitness

### C.1    Ablation study: Comparison between unconditional models and matched generative models

Table 3 and Table 4 below show empirically the advantage of our proposed method. In this experiment, the architecture and training/sampling hyperparameters are the same for both the unconditional and the matched models (the only difference being the matched conditioning on the latter).

---

[9]We use the official implementation provided by the authors in `https://github.com/NVlabs/edm2`.

[10]We use the codebase provided by the authors: `https://github.com/facebookresearch/flow_matching`.

|  | Medium difficulty | | | Hard difficulty | | |
|---|---|---|---|---|---|---|
| Method | Fitness | Diversity | Novelty | Fitness | Diversity | Novelty |
| VAE | 0.39 (0.01) | 12.2 (0.3) | 5.3 (0.6) | 0.30 (0.02) | 14.8 (0.4) | 5.0 (0.0) |
| mVAE | 0.48 (0.02) | 9.5 (0.3) | 6.0 (0.0) | 0.38 (0.04) | 12.0 (1.2) | 7.3 (0.8) |
| FM | 0.35 (0.00) | 14.4 (0.4) | 6.2 (0.5) | 0.27 (0.00) | 16.8 (0.2) | 8.5 (0.5) |
| mFM | 0.52 (0.01) | 6.2 (0.2) | 5.6 (0.6) | 0.35 (0.02) | 6.6 (0.3) | 5.2 (0.5) |
| WJS | 0.37 (0.01) | 14.1 (0.3) | 6.8 (0.5) | 0.28 (0.00) | 17.3 (0.3) | 9.1 (0.3) |
| mWJS | 0.53 (0.01) | 5.2 (0.2) | 5.6 (0.6) | 0.54 (0.04) | 4.6 (0.7) | 6.6 (0.5) |

Table 3: Comparison between (unconditional) models and matched generative models on AAV benchmarks results. Results are shown with mean/standard deviation across 5 runs.

|  | Medium difficulty | | | Hard difficulty | | |
|---|---|---|---|---|---|---|
| Method | Fitness | Diversity | Novelty | Fitness | Diversity | Novelty |
| VAE | 0.74 (0.05) | 1.3 (0.1) | 6.3 (0.6) | 0.31 (0.02) | 6.2 (1.4) | 11.6 (1.5) |
| mVAE | 0.84 (0.03) | 1.9 (0.2) | 7.0 (0.7) | 0.78 (0.04) | 1.3 (0.2) | 7.5 (0.6) |
| FM | 0.14 (0.01) | 11.5 (0.2) | 9.2 (0.4) | 0.08 (0.01) | 13.3 (0.4) | 11.4 (0.6) |
| mFM | 0.50 (0.03) | 5.3 (0.2) | 7.0 (0.0) | 0.55 (0.04) | 5.4 (0.1) | 7.7 (0.5) |
| WJS | -0.02 (0.01) | 83.5 (2.2) | 63.7 (5.7) | -0.06 (0.02) | 80.2 (3.0) | 61.5 (1.5) |
| mWJS | 0.76 (0.03) | 3.2 (0.1) | 6.0 (0.0) | 0.78 (0.02) | 2.9 (0.2) | 7.0 (0.0) |

Table 4: Comparison between (unconditional) models and matched generative models on GFP benchmarks results. Results are shown with mean/standard deviation across 5 runs.

## C.2 Ablation study: influence of the constraint criterion

Table 5 shows ablation studies on the $\Delta_x$ constraint (Levenshtein distance) for the mWJS model on AAV medium task. We observe that fitness remains relatively stable with lower $\Delta_x$ thresholds (0.50-0.53 for $\Delta_x$ in 1-7), peaking at 0.53 for $\Delta_x$ values of 4, 5, and 6. As $\Delta_x$ increases beyond 7, fitness notably declines, reaching 0.45 at $\Delta_x$=10. Diversity generally increases with $\Delta_x$, while novelty shows minor fluctuations. These results indicate that a moderate range for $\Delta_x$ (e.g., 4-7) yields optimal or near-optimal fitness. Very small thresholds likely overly restrict generation, while very large thresholds may dilute meaningful local relationships.

| $\Delta_x$ | 1 | 2 | 3 | 4 | 5 | 6 | 7 | 8 | 9 | 10 |
|---|---|---|---|---|---|---|---|---|---|---|
| Fitness | .52 | .50 | .52 | .53 | .53 | .53 | .52 | .48 | .46 | .45 |
| Diversity | 4.3 | 6.0 | 6.0 | 5.7 | 5.2 | 5.3 | 5.9 | 6.8 | 7.3 | 8.2 |
| Novelty | 5.5 | 5.5 | 4.6 | 5.5 | 5.5 | 5.5 | 4.6 | 5.5 | 5.0 | 5.5 |

Table 5: Ablation study on the effect of $\Delta_x$ threshold (while keeping $\Delta_g$ fixed at .2) on AAV medium for the mWJS model.

Table 6 shows ablation studies on the $\Delta_g$ constraint for the mWJS model on AAV medium task. Here, we observe that performance is stable for smaller $\Delta_g$ values. However, fitness declines when $\Delta_g$ exceeds 0.8.

| $\Delta_g$ | .1 | .2 | .4 | .8 | 1.0 | 1.2 |
|---|---|---|---|---|---|---|
| Fitness | .54 | .53 | .55 | .51 | .51 | .51 |
| Diversity | 5.5 | 5.2 | 5.5 | 5.7 | 5.3 | 4.9 |
| Novelty | 4.7 | 5.6 | 4.7 | 4.7 | 4.7 | 4.7 |

Table 6: Ablation study on the effect of $\Delta_g$ threshold (while keeping $\Delta_x$ fixed at 5) on AAV medium task.

## C.3 Ablation study: choice of initial seeds

In our main experiments, we start the iterative sampling with the top 128 seeds in the train set. Table 7 compares performance of the mWJS variant when selecting when starting with the top seeds versus randomly choosing the initial seeds (independent of their property value).

| Method | Medium difficulty | | | Hard difficulty | | |
|---|---|---|---|---|---|---|
| | Fitness | Diversity | Novelty | Fitness | Diversity | Novelty |
| fixed top seeds | 0.53 (0.01) | 5.2 (0.2) | 5.6 (0.6) | 0.54 (0.04) | 4.6 (0.7) | 6.6 (0.5) |
| random seeds | 0.52 (0.02) | 5.1 (0.3) | 5.8 (0.4) | 0.53 (0.07) | 3.8 (0.7) | 7.0 (0.4) |

Table 7: Comparison between fixed (5 runs) vs random initial seeds (100 runs) for AAV medium and hard tasks (mean/std reported).

We observe that the performance across all metrics (fitness, diversity, and novelty) remains consistent, regardless of whether fixed or random seeds are used. The slight variations observed are well within the reported standard deviations, confirming the stability of the results.

## C.4 Protein fitness vs number of iterations plots

Figure 8 shows plots of protein fitness improvement vs. number of iterations starting with different initial seeds and generating 200 designs per iteration.

# D Additional results on iterative training on self-generated data

## D.1 Ablation study

We performed an additional ablation, comparing our original GPE method to a version where the selected designs for each round are conditioned on a discriminator.

**Experimental Setup.** We trained an ensemble of 15 discriminators, each sharing the mVAE encoder architecture with an added classification head. These discriminators were trained to predict improvement labels, $\mathbb{I}(x,x')$. We used the *charge at pH7* property for consistency with section 4.2. In our first baseline (GPE + Discriminator), we filtered GPE-generated designs, $x'$, using the mean predicted probability from the ensemble and retaining only designs with low prediction variance ($< 0.05$). For an upper bound on performance, we also created a second baseline (GPE + Oracle) where we filtered generated designs using the ground-truth oracle.

**Results.** Our findings in Table 8 show that the discriminator-based filtering scheme did not improve performance. In fact, it underperformed our original GPE method, while the oracle-filtered version achieved a slight early-round improvement. In the table below, RI and AI are the "Ratio of Improvement" and "Average Improvement".

| Method | Round 1 (RI / AI) | Round 2 (RI / AI) | Round 3 (RI / AI) |
|---|---|---|---|
| GPE + Discriminator | $0.57\pm0.022/0.83\pm0.018$ | $0.67/0.92$ | $0.67\pm0.017/0.90$ |
| GPE + Oracle | $0.63\pm0.019/1.01$ | $0.79\pm0.015/1.51$ | $0.80\pm0.014/1.53$ |
| GPE (ours) | $0.60\pm0.023/0.98$ | $0.76\pm0.017/1.45$ | $0.78\pm0.016/1.51$ |

Table 8: Performance comparison of GPE variants across rounds. RI = Ratio of Improvement; AI = Average Improvement.

We attribute the discriminator's poor performance to domain shift. The discriminator was trained on $(x,x')$ pairs with at most two Levenshtein edits, matching the constraints used to build the training set. However, GPE-generated pairs in later rounds span one to ten edits, making them out-of-distribution for the discriminator. This is reflected in the metrics presented in Table 9.

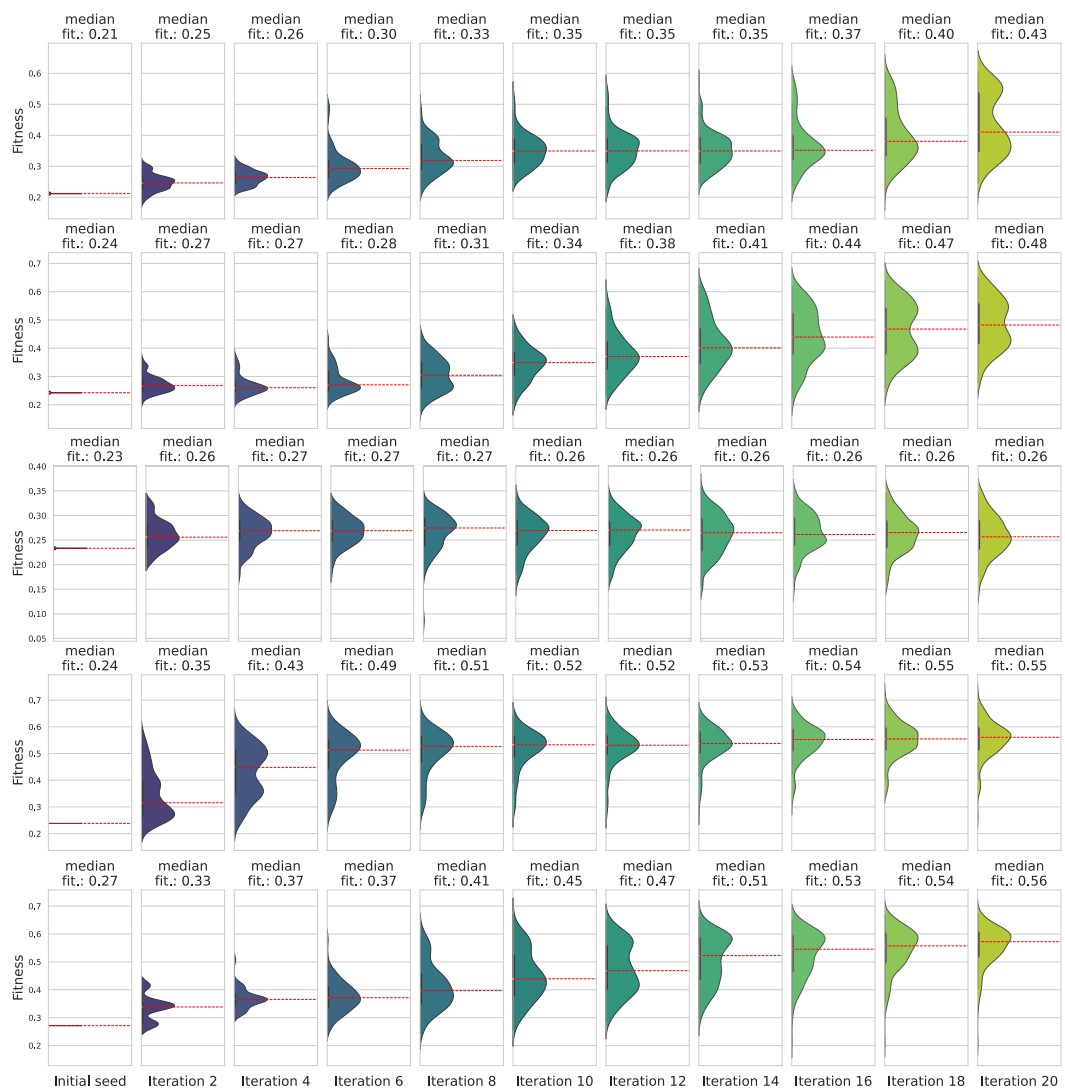

Figure 8: Protein fitness improvement as number of iterations increase. We start from single random seed and use the mWJS model trained on AAV hard to sample designs. The plots show the distribution of fitness of designs for a different initial seeds not seen during training (rows) at different sampling iterations (columns). The first column show the fitness of the initial seed.

| Evaluation Set | Accuracy | Precision | Recall | Variance |
|---|---|---|---|---|
| On 10% Holdout Set (IID) | $0.94\pm0.005$ | $1.00\pm0.001$ | $0.93\pm0.008$ | 0.007 |
| On GPE Designs | $0.659\pm0.021$ | $1.00\pm0.002$ | $0.659\pm0.026$ | 0.021 |

Table 9: Discriminator performance on IID and out-of-distribution (GPE) data.

## D.2 Diversity analysis

Not all seeds are equal. For some which have more neighbors, i.e. lie in a more dense region in the training data, the model has better chances of learning the implicit direction of improvement and therefore more opportunity for generating diverse candidate designs. Hence, the number of pairs around each seed at round 0, is indicative of the opportunities of improvement.

In Table 10, we notice the expected trend, that is, at each round the number of designs (diversity) per seed goes down both in terms of mean and standard deviation.

| Statistic | Round 1 | Round 2 | Round 3 |
|---|---|---|---|
| Count | 3,554 | 4,054 | 4,319 |
| Mean | 3.43 | 2.06 | 1.71 |
| Std | 14.28 | 7.63 | 5.86 |
| Min | 1.00 | 1.00 | 1.00 |
| Max | 101.00 | 101.00 | 101.00 |

Table 10: Descriptive statistics of GPE-generated designs across rounds.

# E   Wet-lab validation

As additional empirical evidence we provide a summary of wet-lab experiments comparing GPE (mPropEn variant) against unconditional WJS [30] for improving therapeutic protein binding affinity across eight initial seeds over three distinct targets as presented in [17]. Our model demonstrates significantly higher and more consistent binding rates (94.6% vs. 62.2%) and design improvements (34.4% vs. 4.8%). Additional experiments benchmarking mVAE vs mPropEn are in progress and will be added to the manuscript as soon as available.

# F   Qualitative results on rotating MNIST with mWJS

To showcase that our model also works for continuous representation, we test our model on a simple task derived from MNIST dataset. In this toy task, we artificially create a property by rotating digits clockwise. The matched dataset is defined such that for each digit in the dataset, we apply two rotations and give the highest property to the digit rotated with highest angle, i.e., given two random angles $\theta_1$ and $\theta_2$ s.t. $\theta_1 - \theta_2 < 30°$ and $\theta_1 > \theta_2$, then $g(R(x,\theta_1)) > g(R(x,\theta_2))$.

We train a matched walk-jump sampling model (using a tiny unet model as denoiser) on this dataset and observe qualitatively what we would expect: starting from an unseen digit, the model samples digits are increasingly rotated clockwise (that is, with higher property).

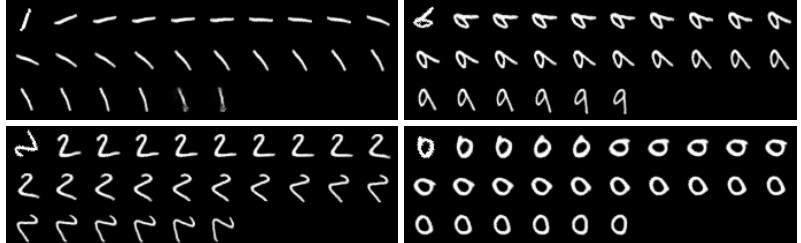

Figure 9: Qualitative results on rotating MNIST. Given an unseen digit (the first on the sequence), our model sample digits that have higher property (i.e. are more clockwise rotated). Each figure is a single MCMC chain generated by the masked walk-sampling model.

