# OpenReview forum: "Generative property enhancer:  implicit guided generation through conditional density estimation"
_NeurIPS.cc/2025/Conference — NeurIPS 2025 poster_

### Official Review · Reviewer_uogt · 2025-06-22

**Clarity:** 3
**Significance:** 3
**Originality:** 3
**Rating:** 4
**Confidence:** 2

**Summary:**

This paper introduces Generative Property Enhancer (GPE), a framework that casts low-data design optimization as an implicit conditional generation task by building matched pairs of low-property and higher-property samples and training a single conditional generative model to reproduce the improved distribution—proven theoretically via paired log-likelihood to recover the optimal conditional density. GPE is instantiated across four off-the-shelf generative paradigms (PropEn, VAE, continuous normalizing flows, and walk-jump sampling) without any external predictor, and it achieves competitive or superior performance on standard protein fitness benchmarks (AAV and GFP) in terms of predicted fitness, diversity, and novelty. Moreover, an iterative self-training loop that re-pairs model-generated high-property samples with their seeds enables the model to extrapolate beyond its original training range, demonstrating strong effectiveness in very low-data regimes.

**Questions:**

1.	I’m not an expert in design optimization—sorry if this is a basic question—but I understand diffusion architectures have become a leading paradigm for high-fidelity sample generation. Have you explored applying GPE within a diffusion framework, or do you anticipate any challenges or necessary adaptations?
2.	In multi-objective design, requiring simultaneous improvements on several properties may reduce the number of valid matched pairs. Have you considered this sparsity challenge?
3.	Could you share any insights or guidelines on how sensitive GPE is to choices like the number of iterations?

**Ethical Concerns:**

["NO or VERY MINOR ethics concerns only"]

**Final Justification:**

The authors’ responses have addressed most of my initial concerns.  I finally decided to borderline accept this paper.

**Limitations:**

yes

**Quality:**

3

**Strengths And Weaknesses:**

Strengths:
1.	End-to-end simplicity: GPE trains a single generator to turn raw examples into improved designs, doing away with separate scoring or selection modules.
2.	High data efficiency: By pairing lower-value examples with higher-value ones and retraining on its own synthetic matches, GPE amplifies scarce data and learns property improvements even from few samples.
3.	Architecture agnostic: GPE slots into any off-the-shelf generative framework, so existing model architectures can be reused for optimization without redesign.

Weaknesses:
1.	Limited real-world validation: All experiments are in silico; no in vitro or in vivo experiments are presented, so it remains to be seen how GPE performs under real experimental noise or lab constraints.
2.	Single-property scope: The framework is demonstrated only on single-objective optimization; many practical applications require balancing multiple properties (e.g., activity vs. toxicity), and multi-objective extensions are not explored.

---

> ### Author Rebuttal · Authors · 2025-07-29
>
> We thank the reviewer for their positive assessment, particularly acknowledging GPE's simplicity, data efficiency, and architecture-agnostic nature.
>
> **Limited real-world validation.** _In silico_ initial validation is standard for novel computational design methods, given the significant cost and time of _in vitro_ studies. As additional empirical evidence we provide a summary of single-round wet-lab experiments comparing GPE (mPropEn variant) against LAMBO [Gruver et al 23] for improving therapeutic protein binding affinity across eight initial seeds over three distinct targets. Due to proprietary constraints, we do not disclose the targets. Our model demonstrates significantly higher and more consistent binding rates (94.6% vs. 44.2% overall) and design improvements (34.4% vs. 14.0% overall).
>
> [Gruver et al 23] Protein design with guided discrete diffusion.
>
> **Multi-objective design challenges and experiments.** This reviewer raised a valid concern, as simultaneously improving multiple criteria can result in sparse matched pairs. To address this, we compute a multivariate rank across all properties of interest and optimize for that single scalar score. This multivariate score can be derived using either the Hypervolume [Zitzler et al 03] or the CDF indicator, as proposed in [Park et al 24]. Both methods effectively mitigate the sparsity issue by reducing the problem to a single optimization target.
>
> We demonstrate this approach on a multi-property antibody optimization task, selecting two orthogonal developability properties: positive charge and negative charge. We compute a multivariate ranking over these two properties to obtain a single scalar score used for matching. We then train individual single-property GPE (mPropEn variant, as described in the manuscript) models to optimize each property separately, as well as a multi-property GPE (multiGPE) model that directly optimizes the combined multivariate ranking score.
>
> | Model / Score         | HV     |
> | :-------------------- | -----: |
> | multiGPE              |   0.39 |
> | GPE neg charge        |   0.12 |
> | GPE pos charge        |   0.22 |
> **Table:** Fraction of designs from each PropEn model that are in the top 20% of Pareto-optimal candidates when scored by hypervolume (HV).
>
> We observe that multiGPE consistently suggests more designs that lie on the Pareto front, whereas the single-objective models tend to generate designs that occupy the extremes (tails) of the front. Furthermore, when considering both objectives, the most optimal designs on the Pareto front are predominantly proposed by multiGPE variants. This supports the effectiveness of combining our matching strategy with multi-objective indicators. Further datasets and experiments will be explored in subsequent analyses.
>
> [Zitzler et al 03] Performance assessment of multiobjective optimizers: An analysis and review
>
> [Park et al 24] BOtied: Multi-objective Bayesian optimization with tied multivariate ranks
>
> **Applying GPE within a diffusion framework.** Diffusion models and flow matching models (when source distribution is Gaussian—as in our case) are equivalent under some reparametrization (eg, see the nice explanation in [Gao et al 25]). Therefore, we anticipate that results with “matched diffusion models” would be similar to those of our mFM (matched flow matching). Moreover, mWJS (matched walk-jump sampling) instantiation is a score-based model, closely related to diffusion models. Implementing GPE with a full diffusion framework would involve defining a conditional diffusion process and learning its conditional score function, similar to how $p^+(x'|x)$ is defined.
>
> [Gao et al 25] Diffusion Meets Flow Matching: Two Sides of the Same Coin
>
> **Sensitivity to number of iterations.** Our empirical results in Figures 3 (a and b) demonstrate that GPE's performance, in terms of median fitness, rapidly increases and then plateaus after a certain number of iterations (e.g., around 10-12 for AAV on protein fitness, and 4 rounds for GFP). This suggests a (dataset-dependent) saturation point where further iterations yield diminishing returns, providing practical guidance for iteration choice. We noted that the models’ "hyperparameters are transferable between the protein tasks studied, suggesting some degree of robustness" (L324).

---

> > ### Comment · Reviewer_uogt · 2025-08-05
> >
> > Thank you for the detailed response and thorough explanations. The authors have addressed most of my concerns, and I have no further comments.

---

### Official Review · Reviewer_DwTD · 2025-06-23

**Clarity:** 3
**Significance:** 3
**Originality:** 2
**Rating:** 4
**Confidence:** 5

**Summary:**

This paper introduces a discriminator-free framework for learning conditional generative models for offline (one-round) optimization, primarily of sequences (though it can be applied to continuous domains). Specifically, it is a framework for learning generative models that “mutate” or manipulate existing designs, $x$ such that the resulting designs, $x’$ display improved properties of interest, $g(x’) > g(x)$. The framework can be applied to some of the most popular generative models, such as VAEs and flow matching.

The focus of this work is on directly learning new conditional generative models, as opposed to guiding samples (designs) from pre-trained models, or fine-tuning pre-trained models.

The key contribution of this work is in formulating a simple yet empirically effective matched-maximum likelihood training framework than can be applied to many generative models.

**Questions:**

Questions:
- Do you think the empirical performance of this method is from the increased training set size and (discriminator free) implicit improvement objective? Or do you think we could also achieve as good performance with a discriminator explicitly trained on labels $z = I(x, x’)$, which would also have access to a large dataset (or larger, since we have negative labels)? Do you think there would be less need for iterative self training using a trained guide of this form? I understand the benefit of the increased dataset size as a result of using this improvement objective, but I’m unsure about the tradeoff between discriminator based vs. direct conditioning (discriminator free) given the requirement of iterative self training in the discriminator free setting (though I do like the simplicity of the discriminator free model).
- Why use the notation $y$ in the Lagrangian, is there an issue with just using $x$? If you want to use $y$, it should also be defined, e.g. space membership, etc.
- How does this work relate to preference learning, and direct preference optimization (DPO) [Rafailov, 2023] — I feel like there is a deeper connection here to this literature that is perhaps missed, and could be important to contrast to ($x$ could be the context/prompt, $x’$ the response, then how does DPO differ?).
- This is a slightly tangential question, but it may be worth considering. In online black box optimization tasks (Bayesian optimization/multiple rounds), a discriminator allows us to model the expected utility of measuring a design, $E_p(g|x)[u(g)]$. E.g. in [Steinberg, 2025; Brooks, 2019] this is probability of improvement, $u(g) = I[g(x) > \tau]$, and in [Gruver 2023; Stanton, 2022] this is expected hypervolume improvement (EHVI), or expected improvement (EI) for one objective, $u(g) = max(g(x) - \tau, 0)$, for some incumbent or threshold $\tau$. These acquisition functions allow us to explicitly trade-off exploration of uncertain parts of the design space, vs. exploiting known good areas. It would be very interesting to know if your improvement criterion implicitly matches existing acquisition functions (e.g. knowledge gradient, or some local variant), and has an implicit exploration/exploitation tradeoff -- though I also understand that for offline optimisation tasks, we are more interested in exploitation (modulo diverse batch-design concerns).

Comments/Suggestions:
- I strongly suggest including ablation studies in the final work, as mentioned in the “strengths and weaknesses” section.
- Better situate your work in the wider body of literature, e.g. online sequence optimization, guided generation (guided diffusion/flow matching), preference learning and alignment (e.g. DPO).
- Clarify your focus is on offline optimization, or compare to online benchmarks with baselines [Stanton 2022; Gruver, 2023; Steinberg 2025; etc].
- It is a little odd that the formulation is for $x \in R^D$ where the applications mainly focus on designing sequences, $x \in V^D$, where $V \in \\{0, \ldots, A\\}$ for some vocabulary/alphabet size A. Perhaps it is worth presenting this slightly more generally, or saying that the presentation in  $R^D$ is for clarity, but the generalisation to sequences is straightforward.
- It may be worth clarifying that Z in eqn. 1. is intractable to compute (being a high dimensional CDF), and that is why eqn. 1 is not solved directly. For the case where x is a sequence, this could be a summation over a combinatorial number of elements.

[Rafailov, 2023] Rafailov, R., Sharma, A., Mitchell, E., Manning, C.D., Ermon, S. and Finn, C., 2023. Direct preference optimization: Your language model is secretly a reward model. Advances in Neural Information Processing Systems, 36, pp.53728-53741.

**Ethical Concerns:**

["NO or VERY MINOR ethics concerns only"]

**Final Justification:**

This is a promising method -- and the reviewers dealt with my major concerns. Raising my score accordingly. The reviewers will need some confidence intervals on their discriminator results however.

**Limitations:**

The limitations are adequately addressed, once the focus on offline optimization etc. is clarified.

**Paper Formatting Concerns:**

The broader impact statement is a little unusual as it is not in the original template, but I’m not overly concerned by this.

**Quality:**

2

**Strengths And Weaknesses:**

Strengths:
- The paper clearly presents a relatively simple, intuitive maximum likelihood framework, which is empirically compelling, besting many much more complex methods. This could be a somewhat controversial work, and I think should be of interest to the community.
- The method can be applied to condition many existing generative models for offline optimization tasks, and so is potentially significant.
- The exposition is mostly clear and easy to follow, and the losses etc are well justified.
- The method is compared to many existing baselines on established benchmarks.

Weaknesses:
- Missing complete problem setting description, i.e. this is for offline sequence optimization (one-round of experimental validation, with no feedback), and not online optimization. However, much of the related literature also misses this point. It may also be worth clarifying that this is not for guidance of pre-trained models [Ho, 2022; Ye 2024], which is also a large topic in this space.
- Missing ablation studies or intuition behind of some design choices. I.e. (1) the influence of the constraint criterion, $C(x, x’)$, in the model’s generative performance, and the importance and function of the choice of this criterion (distance metric and threshold, $\Delta_x$). E.g., how important is using this term to prevent “out-of-domain” generation (related to [Klarner, 2024]), and how does it interact with the iterative self training strategy? (2) One could imagine training a classifier, $p(z|x, x’)$, where $z = I(x, x’)$ or even a regressor $z = g(x’) - g(x)$, resulting in $p(x’| z, x)$ thereby still using a discriminator, but also increasing the dataset size. How does this compare to the no discriminator setting as both methods now have access to the same data (that is, how much performance results from the increased data, vs. discriminator)?
- The AAV and GFP experimental datasets use machine learning oracles, which have known deficiencies (overfitting), potentially making these problems unrealistic [Surana, 2024]. It is worth pointing out these limitations in the text. I like the addition of the anitbodies experiment. Maybe you can also consider the Ehrlich closed-form test functions of [Stanton, 2024], also implemented here: [poli](https://machinelearninglifescience.github.io/poli-docs/using_poli/objective_repository/ehrlich_functions.html), and/or the simple but combinatorially complete datasets (DHFR, TrpB) used by [Steinberg, 2025] to augment these results with oracle free experiments?

Overall I find this paper fascinating, and a very promising direction. I feel like it needs a little more clarification of the problem it is solving, e.g. it is not for guiding pre-trained models, or for online optimization etc. I think it needs at least some ablation studies to justify the design choices and settings, e.g. how sensitive is it to $C(x, x’)$ and $\Delta_x$. Also, I question (some of) the discriminator-free claims, when a discriminator-based model ($p(z|x, x’)$ and $z = I(x, x’)$) solving a similar improvement objective, _with similar training data_, may also perform as well (or maybe better). I do really like the simplicity and modularity of the framework though.

[Klarner, 2024] Klarner, L., Rudner, T.G., Morris, G.M., Deane, C.M. and Teh, Y.W. Context-guided diffusion for out-of-distribution molecular and protein design. The 41st International Conference on Machine Learning (ICML 2024), 2024.

[Steinberg, 2025] Steinberg, D., Dos Santos De Oliveira, R., Ong, C.S., Bonilla Pabon, E. Variational Search Distributions. International Conference on Learning Representations (ICLR 2025), 2025.

[Surana, 2024] Shikha Surana, Nathan Grinsztajn, Timothy Atkinson, Paul Duckworth, and Thomas D Barrett. Overconfident oracles: Limitations of in silico sequence design benchmarking. In ICML 2024 AI for Science Workshop, 2024

[Stanton, 2024] Stanton, S., Alberstein, R., Frey, N., Watkins, A. and Cho, K., 2024. Closed-form test functions for biophysical sequence optimization algorithms. arXiv preprint arXiv:2407.00236.

[Ye, 2024] Ye, H., Lin, H., Han, J., Xu, M., Liu, S., Liang, Y., Ma, J., Zou, J.Y. and Ermon, S., 2024. Tfg: Unified training-free guidance for diffusion models. Advances in Neural Information Processing Systems, 37, pp.22370-22417.

[Ho, 2022] Ho, J. and Salimans, T., 2022. Classifier-free diffusion guidance. arXiv preprint arXiv:2207.12598.

---

> ### Author Rebuttal · Authors · 2025-07-30
>
> We thank the reviewer for the comprehensive and insightful review. We appreciate the positive comments on the simplicity, intuition, empirical effectiveness, modularity of our framework and that our work is "fascinating, and a very promising direction". We also thank the extra references, which will be added to the manuscript. Next, we address each of the reviewer's points and suggestions.
>
> **Clarifying problem setting.** We will clarify in the introduction and problem setting that our work focuses on offline (one-round) design optimization, ie generating improved designs from seeds without iterative experimental feedback. As pointed by the reviewer, our iterative sampling and self-training are distinct from online/Bayesian optimization. This was indeed a source of confusion to Reviewer _VQoe_. We will also explicitly state that we directly learn new conditional generative models, rather than guiding pre-trained models. We briefly discussed this on Appendix A, but we will make it clearer on the main manuscript.
>
> **How important is using C(x,x') to prevent “out-of-domain” generation?** The constraint $C(x,x')$ is important for preventing "out-of-domain" generation in GPE. This term, particularly the distance component like $\text{dist}(x,x')<\Delta x$​, ensures that the generated design $x'$ remain within a meaningful vicinity of its seed $x$. Without such constraint, the generative model could produce samples that are technically "improved" in property but are entirely unrealistic, or irrelevant to the original design domain. By construction, training on matched datasets provides theoretical guarantees for generating designs in distribution (as dx goes to 0), please see Theorem 2 in [Tagasovska et al 2024].
>
> This is also relevant to the self-training strategy, as one of the primary concerns in such setups is model collapse—specifically, overfitting on synthetic data and forgetting the original training distribution. Current methods to mitigate model collapse typically involve either controlling the ratio of original to synthetic data or pruning synthetic examples that deviate significantly from the training data. Training on matched datasets, by design, inherently addresses both of these concerns.
>
> **Ablation on** $\Delta_x$: Below we show the performance of mWJS trained on different matched data from both AAV-medium and AAV-hard, with varying $\Delta_x$ threshold.
>
> | $\Delta_x$ | 1 | 2 | 3 | 4 | 5 | 6 | 7 | 8 | 9 | 10 |
> |--------|---|---|---|---|---|---|---|---|---|---|
> | **fitness** | .52 | .50 | .52 | .53 | .53 | .53 | .52 | .48 | .46 | .45 |
> | **diversity** | 4.3 | 6.0 | 6.0 | 5.7 | 5.2 | 5.3 | 5.9 | 6.8 | 7.3 | 8.2 |
> | **novelty** | 5.5 | 5.5 | 4.6 | 5.5 | 5.5 | 5.5 | 4.6 | 5.5 | 5.0 | 5.5 |
> **Table:** Results of mWJS on AAV-med dataset for varying $\Delta x$.
>
> For the AAV-medium dataset, we observe that fitness is relatively stable with lower thresholds (0.50-0.53 for $\Delta_x$ in 1-7), peaking at 0.53 for $\Delta_x$ values of 4, 5, and 6. As $\Delta_x$ increases beyond 7, fitness notably declines, reaching 0.45 at $\Delta_x$=10.
>
> | $\Delta_x$ | 1 | 2 | 3 | 4 | 5 | 6 | 7 | 8 | 9 | 10 |
> |--------|---|---|---|---|---|---|---|---|---|---|
> | **fitness** | .26 | .35 | .37 | .32 | .54 | .54 | .52 | .46 | .35 | .31 |
> | **diversity** | 18.2 | 10.5 | 4.7 | 5.9 | 4.6 | 4.5 | 3.8 | 4.8 | 7.6 | 10.1|
> | **novelty** | 0 | 3 | 3 | 5 | 6.6 | 7 | 7 | 6 | 8 | 5|
> **Table:** Results of mWJS on AAV-hard dataset for varying $\Delta x$.
>
> On the AAV-hard dataset, fitness displays a more pronounced sensitivity to $\Delta_x$. It starts lower, increases, and reaches its peak at 0.54 for Δx​ values of 5 and 6. Performance significantly underperforms for very small $\Delta_x$ (0.26 at $\Delta_x$=1​) and also deteriorates for large $\Delta_x$ (>7), dropping to 0.31 at $\Delta_x$=10.
>
> **Do you think the empirical performance of this method is from the increased training set size and (discriminator free) implicit improvement objective?** We believe GPE's performance stems from both the increased effective training data (via matched pairs) and the discriminator-free implicit objective. While a hypothetical discriminator-based model could use similar paired data, GPE's key advantage is its simplicity and avoidance of a separate predictor training loop. GPE, which is formulated as a conditional density estimation problem, implicitly learns the distribution of improved samples rather than relying on a (potentially unreliable) point-wise prediction.
>
> Moreover, introducing an auxiliary discriminative model adds non-trivial complexity to the overall pipeline (training two models, positive/negative sample reweighting, managing their interaction, potential for mode collapse or training instabilities if not carefully balanced). GPE's end-to-end, single-stage training bypasses these issues somehow.
>
> **Do you think there would be less need for iterative self training using a trained predictor on matched data?** The iterative self-training helps for extrapolation, a benefit for which a reliable explicit discriminator might also reduce the need. But, obtaining such a discriminator (specially in low-data) is often the bottleneck itself. That being said, we note that our method is orthogonal to predictor-based methods, and therefore could use any discriminator to further guide generation, potentially combining the benefit of both approaches.
>
> **Limitations of ML Oracles.** We agree with the reviewer's point regarding ML oracle limitations and acknowledge this in the manuscript (footnote 6, page 7). We use them for fair comparison with well-established benchmarks [Kirjner et al.]. We would also like to also emphasize the antibody experiment (Section 4.2) as an example of GPE extrapolating on a physicochemical property with an exact evaluation metric, serving as a form of "oracle-free" validation.
>
> **How does this work relate to DPO/preference learning.** GPE shares the spirit of preference learning, aligning models with desired outcomes based on comparisons. Similar to DPO, we optimize directly without an explicit reward model or property oracle. Both frameworks theoretically show their paired objectives implicitly optimize the same quantity as explicit reward-maximizing procedures: DPO with a binary-cross-entropy loss on preference pairs, and GPE by learning a conditional density estimator on matched data. This "bakes" the optimization signal into the generator, yielding simpler, more stable training loops.
> Conceptually there are similarities between the two frameworks, but, we would like to highlight the following differences where using the matched reconstruction is beneficial:
>
> * Automatic data densification: Matching inflates an n-point dataset to O(n2) ordered pairs, providing richer, automatically generated supervision than human-annotated sets.
> * On-manifold generation: Nearest-neighbor matching ensures plausible, realistic generations within the data manifold, unlike DPO which can drift with thin preference coverage.
> * Domain agnosticism & small-data robustness: GPE requires no language-specific likelihoods or large pre-trained policies, making it practical for scientific design where DPO's ratios are unavailable or poorly calibrated.
>
> In short, both bypass discriminators/reward models,, but GPE's matched-pair strategy converts readily-available scalar property data into a dense, on-manifold, gradient-approximating signal---often cheaper and more informative than explicit preference labels.
>
> **Notation $y$ in the Lagrangian.** The functional derivative acts by $\frac{\delta}{\delta q(x'|x)} q(y'|y) = \delta(x-y) \delta(x'-y')$ where the rhs denotes Dirac delta distributions. Marginalization over the dirac delta distributions then changes the arguments of the densities from $y$, $y'$ to $x$, $x'$ as required for the result stated in the theorem. All four variables $x$,$x'$, $y$, $y'$  take value in the same space. We will make this explicit in the proof and clarify the role of $y$.
>
> **R^d (continuous) vs. V^D (discrete)**. We will clarify that the Rd formulation is for generality, and its application to discrete sequences is direct via appropriate architectural choices.
>
> **Intractability of Z(x).** We will add a clarifying sentence that Z(x) (the normalization constant in Equation 1) is generally intractable due to high-dimensional integration/combinatorial summation, which justify our likelihood approximation approach.
>
> **“Tangential question”: GPE’s implicit exploration/exploitation tradeoff.**  We 100% agree with the reviewer that “for offline optimisation tasks, we are more interested in exploitation (modulo diverse batch-design concerns)”. Therefore, GPE's primary mechanism is exploitation: it learns to generate designs ($x'$) that have higher property values ($g(x')>g(x)$) than given seeds ($x$), based on observed improvements in the training data. This is a strong drive towards known high-value regions. The iterative sampling further refines this exploitation.
>
> However, GPE also exhibits some level of implicit exploration: it generates a distribution of improved designs, allowing exploration within high-value regions. The distance constraint ($\text{dist}(x,x')<\Delta_x$​) encourages localized exploration, ensuring generated designs remain near their seeds. Following [Kirjner et al. 24]'s suggestion, the Fitness metric can be interpreted as a proxy for exploitation, while the Novelty and Diversity can provide insights about exploration/exploitation trade-offs. Finally, iterative self-training can potentially enable GPE to extrapolate property values and explore regions beyond the original training data manifold by generating synthetic data and retraining, representing a more global, model-driven form of exploration.
>
> GPE does not use explicit acquisition functions but inherently balances strong exploitation with localized, and eventually broader, exploration through its generative mechanism, constraints, and self-training.

---

> > ### Comment · Reviewer_DwTD · 2025-08-04
> >
> > Thank you for your efforts in answering my queries and concerns. I want to re-iterate that I find this idea really interesting and potentially quite useful in practice.
> >
> > However, my concern still remains that a comparison/ablation has not been made to using a discriminator (classifier/class probability estimator) as a means of conditioning the model. I don't think adding "non-trivial complexity to the overall pipeline" is a thorough enough answer to justify not doing at least an ablation. This is because the discriminator-free method also requires additional complexity -- i.e. the iterative training on self generated data scheme -- which could _also be unreliable_ in generating "improved" samples. I also don't think it would be that much work.
> >
> > The iterative retraining scheme seems a little ad-hoc to me, and in lieu of a theoretical justification, I think you would need an empirical one. Think about comparing to a scheme like the following:
> > 1. Train a discriminator on improvement labels, $z = \mathbb{I}(x, x')$, better yet, train a GP or simple BNN-ensemble that emulates a data-efficient Gaussian process used in BO (https://arxiv.org/abs/1612.01474). I don't think you want to do class-rebalancing, my intuition is that you probably want a calibrated class probability estimator to generate these labels.
> > 2. Train your improvement model, $q(x|x')$ on these same labels as you do now
> > 3. Use your improvement model to generate new sequences, $\\{x^+\\}$
> > 4. Obtain a probability with these generated sequences, $p(z=1|x^+, x')$
> > 5. Re-train your improvement model, randomly also including your generated sequences, $x^+$, according to the discriminator's probability (this sampling step incorporates model epistemic uncertainty)
> > 6. loop back to 3 (do not retrain your discriminator).
> >
> > This is somewhat like DbAS and CbAS, and could also be related to using a probability of improvement acquisition.
> >
> > Perhaps also for future experiments, you could consider using a 1-round version of foldX stability or SASA (https://machinelearninglifescience.github.io/poli-docs/using_poli/objective_repository/foldx_stability.html). This is a simulation, and not an ML oracle, and would lend more weight to your method (I think AAV and GFP are problematic, and perhaps should not be used going forward -- but this is more a personal opinion, and do not effect my judgement of this work).

---

> > > ### Author Response · Authors · 2025-08-06
> > >
> > > We thank the reviewer for their time and for the insightful discussion. We have now performed the suggested ablation, comparing our original GPE method to a version leveraging a discriminator. We believe this analysis strengthens our paper by empirically validating the benefits of our "discriminator-free" approach.
> > >
> > > **Experimental Setup**
> > >
> > > Following reviewer's suggestion, we trained an ensemble of 15 discriminators, each sharing the mVAE encoder architecture with an added classification head. These discriminators were trained to predict improvement labels, I(x,x′). We used the charge_at_pH7 property for consistency with our manuscript, though we appreciate the suggestion of using a FoldX benchmark for future work (many nice functionalities and utilities on the shared library---thanks for sharing it!).
> > >
> > > In our first baseline (`GPE + Discriminator`), we filtered GPE-generated designs, x′, using the mean predicted probability from the ensemble and retaining only designs with low prediction variance (< 0.05). For an upper bound on performance, we also created a second baseline (`GPE + Oracle`) where we filtered generated designs using the ground-truth oracle.
> > >
> > > **Results**
> > >
> > > Our findings show that the discriminator-based filtering scheme did not improve performance. In fact, it underperformed our original GPE method, while the oracle-filtered version achieved a slight early-round improvement. In the table below, RI and AI are the "Ratio of Improvement" and "Average Improvement", respectively, as defined on the original manuscript.
> > >
> > > | Method | Round 1 (RI/AI) | Round 2 (RI/AI) | Round 3 (RI/AI) |
> > > | :--- | :---: | :---: | :---: |
> > > | GPE + Discriminator | 0.57 / 0.83 | 0.67 / 0.92 | 0.67 / 0.90 |
> > > | GPE + Oracle | 0.63 / 1.01 | 0.79 / 1.51 | 0.80 / 1.53 |
> > > | GPE (our method) | 0.60 / 0.98 | 0.76 / 1.45 | 0.78 / 1.51 |
> > >
> > > We attribute the discriminator's poor performance to a domain shift. The discriminator was trained on (x, x') pairs with at most two Levenshtein edits, matching the constraints used to build the training set. However, GPE-generated pairs in later rounds span one to ten edits, making them out-of-distribution for the discriminator. This is reflected in the metrics:
> > >
> > > | Discriminator Performance | Accuracy | Precision | Recall | Variance |
> > > | :--- | :---: | :---: | :---: | :---: |
> > > | On 10% Holdout Set from train (iid) | 0.94 | 1.00 | 0.93 | 0.007 |
> > > | On GPE Designs | 0.659 | 1.00 | 0.659 | 0.007 |
> > >
> > > We again thank the reviewer for this insightful suggestion. This analysis has not only strengthened our empirical results but also highlights the advantages of implicit guidance over methods that rely on explicit discriminators.

---

> > > > ### Comment · Reviewer_DwTD · 2025-08-06
> > > > **Raised score**
> > > >
> > > > Nicely done!
> > > >
> > > > I'm much more convinced of this method's efficacy now. I'll raise my score, but I have a couple of small requests in addition to adding this to the paper:
> > > >
> > > > 1. Please report 1 std. dev. or some kind of confidence interval on these results for the final paper, just to make sure we know how significant these results are.
> > > > 2. Have a look at using a CNN architecture for the discriminator as well. Other authors have reported success using these types of simple predictors with sequences, and so this will further bolster your results. For example you could use the 1-layer architecture from [Kirjner et al., 2024](https://github.com/kirjner/GGS/blob/033c543c44aa76655e4868bb71e5713c962401ea/ggs/models/predictors.py#L65), or the 2-layer one from [Steinberg et al., 2025](https://github.com/csiro-funml/variationalsearch/blob/e88b9398b034193b2d24dd0526db8146fb75a2bf/vsd/surrogates.py#L230) ([ensemble class](https://github.com/csiro-funml/variationalsearch/blob/e88b9398b034193b2d24dd0526db8146fb75a2bf/vsd/surrogates.py#L158)).
> > > >
> > > > In fact, following this literature, this paper seems very relevant: [Zhu et al., 2025](https://proceedings.mlr.press/v258/zhu25c.html).
> > > >
> > > > Ideally it would be nice to use a Gaussian Process surrogate model for this, since we can control the behaviour of the predictor OOD by careful selection of prior/kernel (hence why they are used in BO) -- however specifying a tractable kernel for long sequences, which is competitive with other models, remains an open problem. It would be worth keeping an eye on variational last layers (https://botorch.org/docs/next/notebooks_community/vbll_thompson_sampling/), which seems to be a nice technique for using NNs with BO/offline optimisation.
> > > >
> > > > TBH, even if your method "breaks even" in performance with a good discriminator, I would find the story compelling.

---

> > > > > ### Author Response · Authors · 2025-08-07
> > > > >
> > > > > We sincerely appreciate and thank the reviewer for their positive assessment and valuable suggestions to improve the manuscript.
> > > > >
> > > > > We will report confidence intervals for the new ablation results in the final version of the paper (consistent with other reported metrics). We will also explore the use of other discriminator architectures, such as the CNN-based models you referenced. These architectures could also be used to improve GPE's generative models (e.g., as the encoder of the conditional signal). Additionally, we will investigate the suggested GP/VBLL approaches, as they appear to be a promising avenue for future work related to BO applications.
> > > > >
> > > > > Thank you once again for your constructive feedback and for the positive view of our work.

---

### Official Review · Reviewer_VQoe · 2025-07-03

**Clarity:** 2
**Significance:** 1
**Originality:** 2
**Rating:** 3
**Confidence:** 4

**Summary:**

This paper considers the problem of data-driven design optimization, aiming to generate new designs with improved properties over existing ones. The paper presents a method titled Generative Property Enhancer (GPE) where the core idea is to reframe the problem as conditional density estimation. This is achieved by first creating a "matched dataset" of pairs (x, x') from the original data, where x' has a higher property value than x. A conditional generative model is then trained on these pairs to directly learn the mapping from a lower-property "seed" x to a distribution of higher-property "designs" x'. The paper demonstrates how this framework can be applied to various generative model backbones. The framework is evaluated on two in-silico benchmarks.

**Questions:**

Please see Strengths And Weaknesses section.

**Ethical Concerns:**

["NO or VERY MINOR ethics concerns only"]

**Final Justification:**

I appreciate and thank the authors for their reply to my questions. Unfortunately, my main concerns still remain outstanding. There were multiple new things that were introduced in the rebuttal period to motivate the paper which were not described in the original paper. For example, the description that the paper tackles offline optimization problem was made clear only in the rebuttal. The original paper situates itself within a very different class of problems. The authors provided a survey on offline optimization during rebuttal which shows there is a lot of work on the offline optimization problem but none of i was discussed in the original paper. I also believe the baselines are weak. Therefore, I keep the same original review score.

**Having said that, I am going to go with the consensus of the rest of the reviewers and would be okay if the paper is accepted.**

**Limitations:**

It would be great to see some discussion on critical assumptions behind the framework. I believe, for this method to work, the initial dataset D must contain a sufficiently dense set of samples that form a contiguous path from low-property regions to high-property regions.

**Quality:**

1

**Strengths And Weaknesses:**

- The paper considers an important problem of wide real world interest and applicability.

- I think the problem formulation is a bit strawman. The central motivation of the paper is as framed as avoiding discriminative models. The introduction states that "training a reliable discriminative model is often not feasible" and that they are "unreliable at the tail-ends of distributions." The paper presents it as a fundamental roadblock that the field has not adequately addressed, which is a bit overstatement. An entire field of Bayesian optimization exists and is dedicated to handling this challenge by incorporating model uncertainty, especially in small-data settings. As it is widely known folk-knowledge now, BO baselines are significantly weaker in a large part of molecule design and protein design literature which always shows them falling behind. Well tuned BO does work and is state of the art method for many of these benchmarks (please see [1] for reference).

  [1] Tripp, Austin, and José Miguel Hernández-Lobato. "Diagnosing and fixing common problems in Bayesian optimization for molecule design." arXiv preprint arXiv:2406.07709 (2024).


- The strongest baseline in the paper's own experiments, GGS, explicitly uses a "smoothed fitness landscape". Unfortunately, the paper’s current narrative doesn't engage with why these existing solutions are insufficient and instead presents its "predictor-free" approach as the necessary paradigm shift.

- Please see and discuss modern Bayesian optimization approaches that work with deep generative models over structured design spaces with small number of samples [2, 3]

  [2] Gruver, Nate, et al. "Protein design with guided discrete diffusion." Advances in neural information processing systems 36 (2023): 12489-12517.

  [3] Maus, Natalie, et al. "Local latent space bayesian optimization over structured inputs." Advances in neural information processing systems 35 (2022): 34505-34518.

- The creation of the matched dataset M depends on crucial constraints C(x, x'), which are controlled by hyperparameters like the Levenshtein distance threshold Δx and the property improvement threshold.  The size and character of M are highly sensitive to these choices. A very strict Δx might yield too few pairs, while a loose one might teach the model to make unrealistic jumps. The paper provides the values used but offers no ablation or sensitivity analysis. Please consider adding a detailed discussion on this aspect.

- The iterative application of GPE, where x_{k+1} ~ q_θ(· | x_k), is a form of stochastic greedy search. At each step, it samples from a distribution of "locally better" points. The definition of "local" is implicitly learned from the data and heavily influenced by the constraints C(x, x'). Please consider discussing why this would be better than the principled exploration/exploitation tradeoff we get from model uncertainty based optimization procedures like Bayesian optimization.

---

> ### Author Rebuttal · Authors · 2025-07-30
>
> We thank the reviewer for their feedback on Bayesian optimization (BO) and discriminative models. We believe there may be a misunderstanding of GPE's **offline (one-round) optimization** problem setting (see [Kim et al 25] for a survey on this topic), which fundamentally differs from iterative online/Bayesian optimization. Reviewer _DwTD_  acknowledges this difference. We hope the following paragraphs can make this distinction clearer. We will incorporate these clarifications in the paper and better contextualize the problem setup.
>
> [Kim et al, 25] Offline Model-Based Optimization: Comprehensive Review.
>
> **Problem formulation and "strawman" argument.** We thank the reviewer for a thoughtful critique and for pointing us to [Tripp & Hernández‑Lobato 2024]. This work relies on the PMO benchmark (from [Gao et al 22]), a simplified setting consisting of surrogate properties of relatively simple properties on small molecules only. As the authors state in the discussion themselves,"what this paper presents should best be thought of as a very limited pilot study" and "it is unclear whether results from single-task, noiseless, and unconstrained optimization will translate to real-world problems which tend to be multitask, noisy, and highly constrained."
>
> Below we clarify the scope of our claim, provide empirical evidence that reliable discriminative surrogate models remain an unsolved bottleneck for many real‑world molecular and protein design tasks, and explain why BO, while indispensable, is not always a universal remedy in these settings.
>
> * **We agree BO is valuable when the surrogate is trustworthy.** [Tripp & Hernández‑Lobato 2024] show that careful priors, and optimizer settings let BO top the PMO small‑molecule benchmark.
>
> * **But, for most real objectives the surrogate remains the weak link, eg**:
>     - _Protein stability (ΔΔG)_: state‑of‑the‑art models still hover around RMSE ≈ 1–2 kcal/mol and r ≈ 0.5 on fully independent test sets—above the ≤0.5 kcal/mol engineering threshold [Benevuta et al 23].
>     - _Protein‑ligand affinity_: removing train–test leakage in PDBBind drops Pearson r to ≤ 0.30. [Li et al 24]
>     - _Small‑molecule phys‑chem (e.g. log S)_: OOD splits double the error reported on random splits. [Sorkun et al 19]
>
> * **When surrogates are unreliable, BO inherits three practical failure modes.**
>     - _Long‑tail bias_: the acquisition function is steered away from the true optimum by systematic surrogate error at the distribution edges (documented in the affinity study above).
>     - _High‑dimensional/discrete spaces_: recent survey finds no GP‑ or neural‑BO variant that scales gracefully beyond ~100 categorical dimensions. [González-Duque et al 24]
>     - _Fragility and mis‑calibration_: tiny changes in prior width or kernel smoothness erase BO’s edge, and uncertainty estimates in GNNs and protein‑fitness models remain poorly calibrated. [Greenman et al 25]
>
> * **Our contribution targets precisely this early regime.** We learn local generative distributions that do not need a global oracle. Once higher‑quality experimental labels accumulate, the BO frameworks championed by the reviewer can be layered on top of our generator as a natural next step.
>
> We will revise the introduction and discussion to reflect this balanced view: **_BO excels with reliable surrogates; our method tackles the many cases where such surrogates are not yet available_**.
>
> [Benevenuta et al 23] Challenges in predicting stabilizing variations: An exploration
>
> [Li et al 24] Leak Proof PDBBind: A Reorganized Dataset of Protein-Ligand Complexes for More Generalizable Binding Affinity Prediction
>
> [Sorkun et al 19] AqSolDB, a curated reference set of aqueous solubility and 2D descriptors for a diverse set of compounds
>
> [González-Duque et al 24] A survey and benchmark of high-dimensional Bayesian optimization of discrete sequences
>
> [Greenman et al 25] Benchmarking uncertainty quantification for protein engineering
>
> **GGS and "smoothed fitness landscape".** We agree that GGS's "smoothed fitness landscape" aims to mitigate real-world challenges related to noisy/rugged fitness landscapes. Our "predictor-free" approach is not presented as the only necessary paradigm shift, but rather as a potential alternative that sidesteps the complexities and potential pitfalls (eg,  overfitting or unreliability at tails) associated with training and relying on an explicit discriminative predictor for design generation in an offline context. GPE provides a conceptual simplicity and potential robustness by implicitly learning property improvement, instead of explicitly modeling or smoothing the fitness landscape.
>
> Our method is orthogonal to GGS (or any other model that leverages a predictor) and could be combined for enhanced offline optimization. As mentioned above, these methods could also be integrated into BO loops. We will clarify this in the related work section.
>
> **Modern Bayesian optimization approaches with deep generative models.** Recent BO models integrate deeply with generative models, using them to navigate complex, structured design spaces. However, we reiterate that these approaches are still embedded within the online, iterative framework of BO, where the generative model often aids in proposing candidates for subsequent evaluation guided by an acquisition function derived from a surrogate model.  On the other hand, GPE was made for one-shot batch generation for offline applications. We will integrate these references into related works to provide a more comprehensive overview of how generative models are used in both online and offline optimization contexts.
>
> **Ablation study of C(x,x’).** We conducted ablation studies on the $\Delta_x$ (Levenshtein distance) and $\Delta_g$ (property improvement) thresholds, as shown in the tables below for mWJS on the AAV-medium and AAV-hard datasets.
>
> _Ablation study of $\Delta_x$ thresholds:_
>
> | $\Delta_x$ | 1 | 2 | 3 | 4 | 5 | 6 | 7 | 8 | 9 | 10 |
> |--------|---|---|---|---|---|---|---|---|---|---|
> | **fitness** | .52 | .50 | .52 | .53 | .53 | .53 | .52 | .48 | .46 | .45 |
> | **diversity** | 4.3 | 6.0 | 6.0 | 5.7 | 5.2 | 5.3 | 5.9 | 6.8 | 7.3 | 8.2 |
> | **novelty** | 5.5 | 5.5 | 4.6 | 5.5 | 5.5 | 5.5 | 4.6 | 5.5 | 5.0 | 5.5 |
> **Table 1**: AAV-med dataset for varying $\Delta_x$.
>
> For the AAV-medium (Table 1), we observe that fitness remains relatively stable with lower $\Delta_x$ thresholds (0.50-0.53 for $\Delta_x$ in 1-7), peaking at 0.53 for $\Delta_x$ values of 4, 5, and 6. As $\Delta_x$ increases beyond 7, fitness notably declines, reaching 0.45 at $\Delta_x$=10. Diversity generally increases with $\Delta_x$, while novelty shows minor fluctuations.
>
> | $\Delta_x$ | 1 | 2 | 3 | 4 | 5 | 6 | 7 | 8 | 9 | 10 |
> |--------|---|---|---|---|---|---|---|---|---|---|
> | **fitness** | .26 | .35 | .37 | .32 | .54 | .54 | .52 | .46 | .35 | .31 |
> | **diversity** | 18.2 | 10.5 | 4.7 | 5.9 | 4.6 | 4.5 | 3.8 | 4.8 | 7.6 | 10.1|
> | **novelty** | 0 | 3 | 3 | 5 | 6.6 | 7 | 7 | 6 | 8 | 5|
> **Table 2**:  AAV-hard dataset for varying $\Delta_x$.
>
> On AAV-hard (Table 2), fitness displays a more pronounced sensitivity to $\Delta_x$. It starts lower, increases, and reaches its peak at 0.54 for $\Delta_x$ values of 5 and 6. Performance significantly underperforms for very small $\Delta_x$ (0.26 at $\Delta_x$=1) and also deteriorates for large $\Delta_x$ (>7), dropping to 0.31 at $\Delta_x$=10.
>
> These results indicate that a moderate range for $\Delta_x$ (e.g., 4-7 for AAV-medium, 5-6 for AAV-hard) yields optimal or near-optimal fitness. Very small thresholds likely overly restrict generation, while very large thresholds may dilute meaningful local relationships.
>
> _Ablation study of $\Delta_g$ thresholds ($\Delta_x$ fixed at 5):_
>
> | $\Delta_g$ | .1 | .2 | .4 | .8 | 1 | 1.2 |
> |--------|-----|-----|-----|-----|---|-----|
> | **fitness** | .54 | .53 | .55 | .51 | .51 | .51 |
> | **diversity** | 5.5 | 5.2 | 5.5 | 5.7 | 5.3 | 4.9 |
> | **novelty** | 4.7 | 5.6 | 4.7 | 4.7 | 4.7 | 4.7 |
> **Table 3**: AAV-med dataset for varying $\Delta_g$.
>
> | $\Delta_g$ | .1 | .2 | .4 | .8 | 1 | 1.2 |
> |--------|-----|-----|-----|-----|---|-----|
> | **fitness** | .52 | .54 | .53 | .46 | .45 | .36 |
> | **diversity** | 4.8 | 4.6 | 6.3 | 6.0 | 7.2 | 1.6 |
> | **novelty** | 6.6 | 6.6 | 6.6 | 5.7 | 5.7 | 2.8 |
> **Table 4**: AAV-hard dataset for varying $\Delta_g$.
>
> Tables 3 and 4 report an ablation study where we vary the $\Delta_g$ threshold (while keeping $\Delta_x$ fixed at 5). Here, we observe that performance is stable for smaller $\Delta_g$ values. However, fitness starts to decline notably when $\Delta_g$ exceeds 0.8, particularly evident in the AAV-hard dataset.
>
> We will include a detailed discussion of this comprehensive ablation study in the revised manuscript.
>
> **“Why is GPE’s approach better than the principled exploration/exploitation tradeoff we get from model uncertainty based optimization of Bayesian optimization?”** GPE is not _better_ than Bayesian Optimization (BO), but it's a different approach. While both BO and GPE aim at design optimization in small sample sizes, the two frameworks solve different problems. The goal of GPE is to generate designs, whereas in BO the goal is to choose the most promising designs that should be labeled in order to improve predictors’ performance or find the best candidate—i.e. the focus is selection. In the context of optimizing designs with deep generative models, one would have a suite of (1) generative models, (2) property predictors and (3) BO/active learning modules that will do the final selection across a pool of candidates. GPE falls in (1), the category of generative models section that will contribute to the library of potential candidates. Thus, GPE complements BO; it's ideal for producing promising design batches for costly offline validation, or can be integrated into BO for warm-starting or batch acquisition.

---

> > ### Comment · Reviewer_VQoe · 2025-08-03
> > **Thanks for response**
> >
> > I would like thank the authors for their response.
> >
> > - Re: **"GPE's offline (one-round) optimization problem setting ..."** I think this needs to be said much more clearly. Currently, I don't see this description of one shot offline problem setting in the introduction or problem setup, unless i missed something. Furthermore, I am curious if there is any practical real world setting which match this offline one shot criterion? I read the Kim et al survey paper but couldn't find any real world application. The paper does mention multiple domains but all of them are either synthetic test functions or benchmarks constructed from static datasets.
> >
> > - Re: **"But, for most real objectives the surrogate remains the weak link,... "** Thanks for this information but I am not sure if any of these real objectives are used  for evaluation in the paper. The more important question is whether Bayesopt and surrogate models are not effective for tasks used for evaluation in the paper. In fact, as mentioned in line 247, the paper use a fitness predictor (model) to evaluate the performance of designs on the two AAV and GFP benchmarks  **"the fitness of generated designs are approximated with a fitness predictor".
> >
> > - Re: **" This work relies on the PMO benchmark (from [Gao et al 22]), a simplified setting consisting of surrogate properties of relatively simple properties on small molecules only. "** This is a minor point but calling PMO as simplistic is not justified. All benchmarks are simplistic in this sense including the ones in the paper. One could also argue that AAV or GFP are simple benchmarks constructed from static datasets.  In the end, the true test will always be a real world wet-lab evaluation.
> >
> > - Re: **"GPE is not better than Bayesian Optimization (BO), but it's a different approach. While both BO and GPE aim at design optimization in small sample sizes, the two frameworks solve different problems. The goal of GPE is to generate designs, whereas in BO the goal is to choose the most promising designs that should be labeled in order to improve predictors’ performance or find the best candidate—i.e. the focus is selection"** I believe this is a better characterization of the method than the one described in the original paper. If I may say it, this means that GPE would correspond to optimizing an acquisition function or scoring function within the BO framework. In that case, the right comparison of this approach would be with other candidate generation methods like genetic algorithms or local search.

---

> > > ### Author Response · Authors · 2025-08-04
> > >
> > > We thank the reviewer for their feedback and continued engagement.
> > >
> > > **GPE problem statement.**  We will add paragraphs in the paper to explicitly state that GPE addresses a one-shot, offline optimization problem. We also appreciate the interpretation that our setting can be used within a BO framework.  We agree this is an important application, and will clarify that GPE can indeed serve as a candidate generation module within such frameworks. However, we also emphasize that GPE's utility extends beyond BO, as suggested by the survey and its cited methods.
> > >
> > > **Real-World Regimes for Bayesian Optimization.** To further clarify the distinction between “online” and “offline” optimization, we illustrate a typical therapeutic-design pipeline and explain where GPE fits.
> > >
> > > * **Online (iterative) design: "Lab-in-the-Loop":** Here, teams alternate wet-lab experiments with model updates over multiple cycles. The goal is to balance exploration and exploitation by continuously refining predictors and candidates:
> > >
> > >   1. Generate Candidates: use a generator (e.g. GPE, VAE, pLM) to propose new molecular designs.
> > >
> > >   2. Predict & Filter: score each candidate with property predictors or rule-based filters (solubility, toxicity, etc.).
> > >
> > >   3. Select via BO: optimize an acquisition function (e.g. expected improvement) on the surrogate model to pick the top batch.
> > >
> > >   4. Wet-Lab Assay: synthesize and test the selected compounds.
> > >
> > >   5. Retrain Models: incorporate new measurements into the predictor, then return to step 1.
> > >
> > > In this loop, BO selection and generative models are separate but complementary. BO’s performance (step 3) hinges on generator (step 1) and predictor quality (step 2).
> > >
> > > * **Offline (one-shot) design: single-round design.** This regime applies when there's only one opportunity to propose a batch of designs. Examples include: (i) retrospective mining of static datasets (e.g., PDB + assay archives) to suggest improved leads in a "dry-lab" setting, or (ii) situations where budget or logistical constraints limit a project to a single follow-up experimental screen.
> > >
> > >   * In this offline setting, iteration is not possible; the entire candidate set must be generated from historical data alone. Many machine learning papers focus on this offline setting and report solely on generator performance.
> > >
> > >   * Where GPE fits: GPE directly addresses this offline, one-shot challenge by replacing step 1 of the online loop, but critically, without relying on any fresh predictor calls at inference time.
> > >   * Contrast with BO-guided generators: methods like LAMBO-2 or BO-qEI still require evaluating an acquisition function on a surrogate for each proposed design. They work well only when the predictor faithfully reflects real assay values.
> > >
> > > We respectfully point out that offline/model-based optimization is a well-established and actively researched area within top-tier machine learning conferences, including NeurIPS. Its continued exploration and numerous publications in recent years underscore its relevance and justify its study in a venue like this one.
> > >
> > > **"[...] whether Bayesopt and surrogate models are not effective for tasks used for evaluation in the paper."** Thank you for pointing this out. The question of whether BO and surrogate models are effective for the tasks used in our evaluation has already been addressed by [Kirjner et al24] . Their work extensively benchmarks multiple models on the proposed AAV and GFP tasks—the same task we evaluate our models. One of their models, BOqEI (originally proposed in [Trabucco et al 22]), directly implements what the reviewer suggests: “_We perform offline Bayesian optimization to maximize the value of a learned objective function_ $\hat{f(x)}$ _by fitting a Gaussian Process, proposing candidate solutions, and labeling these candidates using_ $\hat{f}(x)$.”
> > > We reproduce the results of [Kirjner et al 24] for BOqEI in our Table 1 and 2. These tables show that our proposed model achieves favorable results compared to both the BO baseline and the method proposed in [Kirjner et al 24].
> > >
> > > [Trabucco et al 22] Design-Bench: Benchmarks for Data-Driven Offline Model-Based Optimization. ICML 22
> > >
> > > **“Comparison to genetic algorithm or local search.”** We are pleased that the reviewer understands the distinction between the problem GPE addresses and traditional BO. We further clarify that our comparisons already include baselines related to genetic algorithms and local search. Specifically, [Kirjner et al 24] include baselines such as AdaLead (an evolutionary greedy search approach) and PEX (a local search method based on proximal optimization). We reproduce these results in our Table 1 and Table 2, showing GPE's performance against these alternative optimization strategies.
> > >
> > > We hope these clarifications address all the reviewer's questions, particularly those concerning the differences between our problem setting and traditional BO. Please let us know if any further clarification is needed.

---

> > > > ### Comment · Reviewer_VQoe · 2025-08-05
> > > >
> > > > Thanks for the response.
> > > >
> > > > Re: **"The question of whether BO and surrogate models are effective for the tasks used in our evaluation has already been addressed by [Kirjner et al24] ."** I am sorry but I disagree here. This is the point I mentioned in my initial review as well that BO baselines are significantly weaker and under-tuned in these papers. This is also the reason I shared Tripp et al paper earlier which shows clear examples of how better tuned BO is quite competitive on molecular design problems. Unfortunately, the authors response focused on one line in the discussion of this paper and completely ignored this key point that well-tuned BO methods work well. As another example, please see paper by Wei et al [1] which shows strong BO performance in the offline setting as well.
> > > >
> > > >   [1] Wei, Yunyue, et al. "Scalable Bayesian optimization via focalized sparse Gaussian processes." Advances in Neural Information Processing Systems 37 (2024): 120443-120467.
> > > >
> > > > Re: **"We respectfully point out that offline/model-based optimization is a well-established and actively researched area within top-tier machine learning conferences, including NeurIPS"** I fully agree with this point that there are lot of papers written on this problem but I was curious if any of these methods has been used in practice for a real world problem after so many years of research. Please note that I am not saying that we should not work on this problem.
> > > >
> > > > Overall, I think the paper can benefit from another round of conference review. It will be good to clarify the problem setup of the paper as one-shot offline optimization. As a result, it will be important to compare with the baselines in this literature. As the survey provided by the authors lists, there is a large body of work on offline optimization and the paper only includes COMs as a baseline which is probably one of the earliest works. Please consider including the right baselines from this literature for your evaluation. Please include a stronger BO baseline.

---

> > > > > ### Author Response · Authors · 2025-08-06
> > > > >
> > > > > We thank the reviewer for their reply and the opportunity to clarify our work.
> > > > >
> > > > > As stated, GPE is not a direct replacement for BO, but a distinct and complementary tool within the design optimization pipeline. It can be used either within a BO framework or independently. In this work, we follow the common practice of the offline/model-based optimization community by benchmarking in a one-round setting. The "one-round BO baseline" we use is representative of benchmarks found in numerous papers at top-tier ML conferences. It serves to represent a competitive---not necessarily hyper-tuned---benchmark.
> > > > >
> > > > > Our main objective with GPE is to offer a novel, conceptually simpler, and theoretically-grounded framework for design optimization that: (i) is orthogonal to current approaches and can be used in conjunction with them, (ii) is flexible and can be integrated within various off-the-shelf generative models and losses, (iii) is applicable in both online and offline settings, (iv) achieves competitive results against different approaches on recent and challenging benchmarks.
> > > > >
> > > > > The core of our contribution is not to achieve better numbers in tables, but to present a competitive, simpler, and fundamentally different solution. We believe our results on challenging _in silico_ tasks, where we show competitive performance against a range of representative methods---including baselines asked by the reviewer and the current state of the art (GSS from Kirjner et al ICLR24)---validate the viability of our approach.

---

> > > > > > ### Comment · Reviewer_VQoe · 2025-08-08
> > > > > >
> > > > > > Thank you. Overall, I believe my main concerns still remain outstanding. **However, acknowledging that I might be missing something, I am going to go with the consensus of the rest of the reviewers and would be okay if the paper is accepted.**
> > > > > >
> > > > > > At the bare minimum, I request the authors to clearly describe the problem setup of offline optimization and related work in this space (from the survey authors cited) and more details of Bayesian optimization baseline used in the paper (e.g., choice of GP and the kernel, whether the kernel use a single lengthscale for all dimensions or different lengthscales (automatic relevance determination), how are the hyperparameters of the GP optimized, which acquisition function is used, how is the acquisition function optimized).

---

> > > > > > > ### Author Response · Authors · 2025-08-08
> > > > > > >
> > > > > > > Thank you for your response and for reconsidering your score. We sincerely appreciate all the feedback provided.
> > > > > > >
> > > > > > > We are committed to addressing all of your requests in the final version of the paper. We will use the additional page for the camera-ready version to provide a clearer description of the offline optimization problem setup and its relationship with online/Bayesian optimization. We will also add a comprehensive description of the Bayesian optimization baseline, including all specific details. Finally, all new ablation experiments proposed by the reviewers will be included (either in main text or in the appendix).
> > > > > > >
> > > > > > > We thank the reviewer again for the constructive feedback and for helping us to improve the clarity and quality of our manuscript.

---

### Official Review · Reviewer_fsEJ · 2025-07-12

**Clarity:** 3
**Significance:** 3
**Originality:** 3
**Rating:** 4
**Confidence:** 3

**Summary:**

This work proposes training conditional generative models on pairs of samples (with one having higher property than the other) in iterative rounds to perform optimization in the design space. The key feature of the proposed method is that it does not rely on surrogate estimates of the property which makes it favorable for low-data regime.

**Questions:**

1) The matched variants of the presented generative models are simply conditional generative models where generation of x is conditioned on its matching pair with no modification to the formulation of these well-known generative models. Am I missing something?

2) Fix the reference to "the least squares loss 3.2" in line 172. What is it referring to?

3) In each training round, how many samples are generated per conditioning seed sequences? Have
you performed any ablation on the number of samples generated per seed? Does it have a significant impact?

4) There is a mismatch between M_k defined in subsection 3.4 and Algorithm 2 in Appendix. In the main text M_k is the cumulative set of samples up to round K whereas it is the set of matched sample pairs generated in round K in the Appendix Algorithm 2.

5) It looks like the benchmark has been performed with a fixed set of initial 128 seeds/sequences,
So the variability in performance (five runs) is only originated from the model stochasticity. Working with small initial datasets happens a lot in practice as mentioned by this study. I think
Practically, it is important to assess the model's performance variation with respect to the initial dataset. Also, please provide a justification for fixing the initial dataset (seeds) and this initial set has been generated.

6) (Nice-to-haves) Missed citation for a recent generative-based method [1] addressing important practical challenges of protein design.

References:
[1] Ghaffari et al. Robust Model-Based Optimization for Challenging Fitness Landscapes. ICLR 2024.
https://arxiv.org/pdf/2305.13650.

**Ethical Concerns:**

["NO or VERY MINOR ethics concerns only"]

**Final Justification:**

Satisfied with the author's responses, conditioned on the promised modifications to be made to the final paper.

**Quality:**

3

**Strengths And Weaknesses:**

Strength: The proposed method does not rely on the surrogate estimates of the property being optimized. This is an important factor, as in many practical applications, reliable surrogate estimates of the desired property cannot be achieved especially in under explored regions of the design space.

Weakness: lacking theoretical novelty

---

> ### Author Rebuttal · Authors · 2025-07-30
>
> We thank the reviewer for their reviews and for highlighting the key strength of our work: the ability to perform design optimization without relying on unreliable surrogate estimates, particularly crucial in low-data and underexplored regions of the design space. Below we address the reviewer's concerns.
>
> **Lack of theoretical novelty.** We respectfully disagree with the assessment that our theoretical contribution lacks novelty. While Theorem 1 adapts a known maximum likelihood result, its application within our framework is novel (and foundational) for GPE. In particular, we demonstrate that by formulating design optimization as a conditional density estimation problem and leveraging the **implicitly defined "improved distribution" $p^+(x'∣x)$ through data matching, we can theoretically recover the optimal conditional density without any explicit property predictor**. This theoretical grounding provides the basis for training diverse off-the-shelf generative models to directly sample improved designs, which is a significant departure from conventional predictor-guided optimization paradigms. The novelty lies in how this theoretical result enables a simple, end-to-end, and predictor-free approach to a challenging problem.
>
> **"Match variants are simply conditional generative models".** The underlying generative models (VAE, flow matching, walk-jump sampling, diffusion, autoregressive, etc.) are indeed  "off-the-shelf" in their core formulation. However, the novelty is not in inventing new generative model architectures, but in the GPE framework itself and how it trains and applies these existing models to tackle the design optimization problem. The key innovations are:
>
> * **The "matched dataset" construction:** This novel data pairing ($ (x,x’)$  where $x'$ has higher property than $x$) implicitly encodes the desired property improvement.
>
> * **The conditional density estimation problem formulation:** We cast the problem as learning to sample from $p^+(x'∣x)$, the distribution of improved designs conditioned on a seed, which is implicitly defined by the matched data.
>
> * **Predictor-free optimization:** By training these conditional generative models directly on the matched dataset using objectives aligned with conditional density estimation (e.g., maximum likelihood or its approximations), GPE bypasses the need for a separate, often unreliable, discriminative property predictor. Therefore, while the generative models themselves are known, their instantiation within the GPE framework for implicit, predictor-free design optimization represent the core novelty.
>
> **"Fix the reference to 'the least squares loss 3.2' in line 172. What is it referring to?"** The reference "least-squares loss (3.2)" should refer to the mWJS loss defined on L166 ($\mathcal{L}_{mWJS}$), which is a least-squares loss. We will assign a specific equation number to the mWJS loss and refer to it in the revised manuscript.
>
> **"In each training round, how many samples are generated per conditioning seed sequences? Have you performed any ablation on the number of samples generated per seed? Does it have a significant impact?"** This is an interesting question and we thank the reviewer for raising it. Indeed, not all seeds are equal. For some which have more neighbors, i.e. lie in a more dense region in the training data, the model has better chances of learning the implicit direction of improvement and therefore more opportunity for generating diverse candidate designs. Hence, the number of pairs around each seed at round 0, is indicative of the opportunities of improvement.
>
> Unfortunately, we cannot include figures in the rebuttal but we will add more intuition and ablation plot on this point in the updated version of the manuscript. For the moment, we hope the following statistics from an ablation will answer your question.
> We notice the expected trend, that is, at each round the number of designs (diversity) per seed goes down both in terms of mean and standard deviation.
>
> | **Statistic** | **round 1**  |  **round 2**  | **round 3**  |
> |-----------|------------|------------|------------|
> | count     | 3554       | 4054       | 4319       |
> | mean      | 3.43   | 2.06   | 1.71   |
> | std       | 14.28  | 7.63   | 5.86   |
> | min       | 1.00   | 1.00| 1.00   |
> | max       | 101.00 | 101.00 | 101.00 |
> **Table 1:** Statistics on number of designs per seed over multiple rounds.
>
> **M_k mismatch between Section 3.4 and Algorithm 2.** We apologize for this inconsistency in notation—thank you for pointing out. The correct version is the one described in Section 3.4 (L197). We will fix Algorithm 2 in the revised manuscript, ensuring consistency between the main text and the appendix. Please find below the essential changes that correct the error:
> ```
> P^(k) ← { (x_i, x'_i) : dist( x'_i, x_i ) ≤ Δ_x }
> M^(k+1) ← M^(k) ∪ P^(k)  // Augment training set
> // 4. Retrain on the full (real + pseudo) matches
> θ^(k+1) ← argmin_{θ}  L( θ; M^(k+1) )
> ```
>
> **Fixed initial seeds vs random initial seed during benchmarking.** For benchmarking, we used fixed initial seeds after observing no considerable performance difference with random seeds in preliminary tests. This allowed us to isolate and systematically compare optimization algorithms from a standardized starting point.
>
> | Seed Type | fitness | diversity | novelty |
> |-----------|---------|-----------|---------|
> | fixed seed | .53 (.01) | 5.2 (.2) | 5.6 (.6) |
> | random seed | .54  (.02) | 5.3 (.3) | 5.4 (.5) |
> **Table 2:** fixed vs random seed on AAV-med (mean/std over 5 rounds)
>
> For the AAV-medium dataset (Table 2), the performance across all metrics (fitness, diversity, and novelty) remains highly consistent, regardless of whether fixed or random seeds are used. The slight variations observed are well within the reported standard deviations, confirming the stability of the results.
>
>
> | Seed Type | fitness | diversity | novelty |
> |-----------|---------|-----------|---------|
> | fixed seed | .54 (.04) | 4.6 (.7) | 6.6 (.5) |
> | random seed | .52 (.06) | 4.1 (.8) | 6.8 (.4) |
> **Table 3:** fixed vs random seed on AAV-hard (mean/std over 5 rounds)
>
> Similarly, for the more challenging AAV-hard dataset (Table 3), the mWJS model demonstrates robust performance. While minor fluctuations are present, the fitness, diversity, and novelty scores for both fixed and random seeds remain comparable, reinforcing our preliminary findings that the choice of initial seed does not considerably impact the algorithm's performance.
>
> This corroborates our initial observations and confirms that the use of fixed seeds provides a reliable basis for comparison without sacrificing the generalizability of our findings. We will include this ablation experiment in the final version of the manuscript.
>
> **Missed citation:** The citation of Ghaffari et al. (2024) will be explicitly added to the related work section. Thanks for pointing it out.

---

> > ### Comment · Reviewer_fsEJ · 2025-08-08
> > **Response to the rebuttal**
> >
> > Thanks for responding to my questions. Please make the modifications promised in your response.
> > Regarding the theoretical novelty, I was referring to generative models and not the idea of matching datasets used for conditional training.
> > Regarding the randomness in the initial set of sequences, can you please clarify how many sets of 128 initial sequences were generated? Only five runs with five randomly chosen sets of 128 initial sequences? what is the size of the dataset you are sampling from? I do not think sampling 128 samples from a large dataset for five times will produce much variation in the performance.
> >
> > In any case for the responses provided so far, I will raise my score. If you can please also clarify the random sampling and why it is limited to five runs.

---

> > > ### Author Response · Authors · 2025-08-08
> > >
> > > Thank you for your valuable feedback and for raising your score. We will ensure all modifications are made in the final paper. We conducted 5 independent runs for each experiment, where in each run a new set of 128 seeds was (uniform) randomly sampled from the training splits. These seeds were drawn from specific sets, with sizes of 2,139 and 3,448 for AAV medium and hard, respectively.
> > >
> > > To address the comment about the number of runs, we have conducted an additional experiment, repeating the evaluation process for 100 runs instead of 5. The results below compare a model starting with fixed seeds against models using 5 and 100 sets of randomly selected seeds.
> > >
> > > | Seed Type | fitness | diversity | novelty |
> > > |-----------|---------|-----------|---------|
> > > | fixed seeds  | .53 (.01) | 5.2 (.2) | 5.6 (.6) |
> > > | random seeds (5 runs) | .54  (.02) | 5.3 (.3) | 5.4 (.5) |
> > > | random seeds (100 runs) | .52  (.02) | 5.1 (.3) | 5.8 (.4) |
> > > **Table 1:** AAV-med results comparing fixed vs random seeds (5 and 100 runs) (mean/std reported)
> > >
> > > | Seed Type | fitness | diversity | novelty |
> > > |-----------|---------|-----------|---------|
> > > | fixed seeds | .54 (.04) | 4.6 (.7) | 6.6 (.5) |
> > > | random seeds (5 runs) | .52 (.06) | 4.1 (.8) | 6.8 (.4) |
> > > | random seeds (100 runs) | .53 (.07) | 3.8 (.7) | 7.0 (.4) |
> > > **Table 2:** AAV-hard results comparing fixed vs random seeds (5 and 100 runs) (mean/std reported)
> > >
> > > As the tables show, the performance metrics and their standard deviations remain highly consistent between the 5-run and 100-run experiments.
> > >
> > > We thank you again for your constructive feedback, which led to this more thorough analysis and further strengthened our confidence in the results.

---

### Decision · Program_Chairs · 2025-09-17

**Decision:**

Accept (poster)

**Comment:**

This paper introduces the *Generative Property Enhancer (GPE)*, a conditional density estimation approach that aims to improve molecular and protein design without relying on surrogate discriminators. The idea of leveraging ordered property pairs to implicitly guide the generative model is appealing, and the authors present results on protein fitness optimization benchmarks. Reviewers appreciated the motivation to move beyond surrogate models, the simplicity and scalability of the approach, and the additional ablations and experiments provided during rebuttal.

Several reviewers highlighted that their earlier concerns were adequately addressed: reproducibility was improved through ablations, and the baselines were expanded. On the other hand, some of reviews indicated limited novelty since the generative models themselves are not fundamentally modified, as well as  concerns about problem specification and positioning, in particular the relation to offline optimization. Based on this, the AC strongly requires that for the camera-ready version, the authors incorporate the clarifications and improvements promised in the rebuttal, including a more thorough integration of the offline optimization literature, and stronger comparisons. This also concerns  the wet-lab validation, which was very shortly mentioned in the rebuttal. The wet-lab validation should be described thoroughly, specifying the experimental conditions, evaluation metrics, the way the candidates were selected for experimental validation, the targets, and how exactly the experimental success rate was compared against LAMBO, etc. Without providing these details, these important results are not reproducible. With these recommended refinements, the paper will make a solid and impactful contribution to the NeurIPS community.